# eIF2A represses cell wall biogenesis gene expression in *Saccharomyces cerevisiae*

**Laura Meyer, Baptiste Courtin, Maïté Gomard, Abdelkader Namane, Emmanuelle Permal, Gwenael Badis, Alain Jacquier, Micheline Fromont-Racine***

Institut Pasteur, Génétique des Interactions Macromoléculaires, Centre National de la Recherche Scientifique, UMR 3525, Paris, France

* mfromont@pasteur.fr

## Abstract

Translation initiation is a complex and highly regulated process that represents an important mechanism, controlling gene expression. eIF2A was proposed as an alternative initiation factor, however, its role and biological targets remain to be discovered. To further gain insight into the function of eIF2A in *Saccharomyces cerevisiae*, we identified mRNAs associated with the eIF2A complex and showed that 24% of the most enriched mRNAs encode proteins related to cell wall biogenesis and maintenance. In agreement with this result, we showed that an *eIF2A* deletion sensitized cells to cell wall damage induced by calcofluor white. *eIF2A* overexpression led to a growth defect, correlated with decreased synthesis of several cell wall proteins. In contrast, no changes were observed in the transcriptome, suggesting that eIF2A controls the expression of cell wall-related proteins at a translational level. The biochemical characterization of the eIF2A complex revealed that it strongly interacts with the RNA binding protein, Ssd1, which is a negative translational regulator, controlling the expression of cell wall-related genes. Interestingly, eIF2A and Ssd1 bind several common mRNA targets and we found that the binding of eIF2A to some targets was mediated by Ssd1. Surprisingly, we further showed that eIF2A is physically and functionally associated with the exonuclease Xrn1 and other mRNA degradation factors, suggesting an additional level of regulation. Altogether, our results highlight new aspects of this complex and redundant fine-tuned regulation of proteins expression related to the cell wall, a structure required to maintain cell shape and rigidity, providing protection against harmful environmental stress.

## Introduction

mRNA translation is a key cellular process composed of four stages: initiation, elongation, termination and ribosome recycling. Protein synthesis is mainly regulated during translation initiation, which mostly occurs through a 5'-Cap-dependent mechanism. The initiation process begins with the formation of a ternary complex between the eukaryotic initiation factor 2 (eIF2, consisting of three subunits α, β, and γ), one molecule of GTP and Met-tRNA. The 43S pre-initiation complex (PIC), composed of the ternary complex, the 40S ribosomal subunit

**Data Availability Statement:** The genomic data were deposited on the ENA Dataset server. The accession number is PRJEB63554 Raw mass spectrometry data were posted in the

ProteomeXchange Consortium via the PRIDE repository. The accession number is PXD043985.

**Funding:** This work was supported by the ANR-17-CE11-0049-01, the ANR-17-CE12-0024-02 and the ANR-18-CE11-0003-04 grants from the Agence Nationale de la Recherche. Financial support and facilities were provided by the Institut Pasteur and the Centre National de la Recherche Scientifique. -MFR, AN, GB and AJ (ANR-17-CE11-0049-01, ANR-17-CE12-0024-02). -AN and AJ (ANR-18-CE11-0003-04). LM and BC were supported by ANR-17-CE11-0049-01, ANR-17-CE12-0024-02 and ANR-18-CE11-0003-04. The funders had no role in study design, data collection and analysis, decision to publish, or preparation of the manuscript.

**Competing interests:** The authors have declared that no competing interests exist.

and different initiation factors (eIF1, eIF1A, eIF5 and eIF3), is recruited on the cap structure at the 5' end of mRNAs. The eIF4F complex consisting of the helicase eIF4A, the cap-binding protein eIF4E and the scaffolding protein eIF4G is then recruited to form the 48S initiation complex (IC). After unwinding of mRNA cap-proximal regions, PIC scans the 5'-UTR until it encounters the initiation codon, which then triggers eIF2-GTP hydrolysis, mediated by eIF5 and the release of initiation factors, including eIF2. The following recruitment of eIF5B strengthens the association of initiator tRNA with the small ribosomal subunit and joining with the 60S ribosomal subunit, resulting in the formation of an 80S ribosome competent for polypeptides synthesis [1–3 and references therein].

Although most mRNAs use this canonical mechanism, translation initiation can also be mediated by alternative cap-independent mechanisms, for example, through internal ribosome entry sites (IRES). This mechanism requires specific, often structured, RNA sequences within 5′ UTRs, that are recognized by the 40S ribosomal subunit to initiate translation, bypassing the scanning process. In mammalian cells, IRES-driven translation was first discovered in viral mRNAs, which are usually uncapped [4, 5]. IRES translation is not limited to viral mRNAs since 10 to 15% of cellular mRNAs can also be translated by this alternative mechanism under stress conditions, such as DNA damage, amino acid starvation or hypoxia, that could impair the canonical translation initiation pathway [6, 7].

Eukaryotic translation initiation is a highly complex process and although enormous progresses have been achieved to elucidate the molecular mechanisms controlling protein synthesis initiation, the function of many factors remains still unclear.

In mammals, in addition to eIF2, there is another factor, eIF2A/IF-M1, which coordinates the binding of the initiator tRNA to the 40S ribosomal subunit. In contrast to eIF2, eIF2A does not require GTP for delivering Met-tRNA to the 40S ribosomal subunit [8, 9]. eIF2 activity is highly regulated by four stress-activated kinases, in response to challenging environments, such as amino acid starvation or virus infection. For example, the "general control non-derepressible 2" kinase (GCN2) phosphorylates the eIF2$\alpha$ subunit, inhibiting eIF2B-mediated nucleotide exchange from eIF2-GDP to eIF2-GTP, thus downregulating eIF2-dependent translation initiation [10 for review]. Translation of a subset of cellular and viral mRNAs is refractory to the inhibitory effects of eIF2$\alpha$ phosphorylation, as shown in the case of hepatitis C viral (HCV) mRNA. During HCV infection, eIF2A coordinates translation of HCV mRNA though its recruitment to the HCV IRES [11]. eIF2A also mediates translation initiation of subgenomic (26S) Sindbis virus mRNA in the absence of functional eIF2 [12]. It has been proposed that under standard conditions, translation initiation is mainly mediated by the canonical pathway *via* eIF2, while eIF2A may be required to ensure persistent translation initiation under stress conditions. However, other did not observe any eIF2A involvement in the translation driven by Hepatitis C virus IRES in human cells [13] or in the translation of Sindbis subgenomic mRNA [14], showing that eIF2A function in translation initiation regulation of viral mRNAs is not yet fully understood. Furthermore, under stress conditions, eIF2A was also required for translation of a non-receptor protein tyrosine kinase c-Src mRNA though its recruitment to the IRES, required for cell proliferation and programmed cell death [15].

Protein synthesis is usually initiated at an AUG, but translation initiation can also occurs at non-AUG start codons, such as AUA, UUG, CUG, ACG or GUG, [16–19]. Furthermore, in response to a range of physiological changes such as accumulation of misfolded proteins, amino acid starvation, viral infection or ER stress, the integrated stress response (ISR) is activated, leading to eIF2$\alpha$ phosphorylation and downregulation of cap-dependent protein synthesis. However, translation of ISR-induced proteins must be maintained, as it was reported for the endoplasmic reticulum chaperone BiP (binding immunoglobulin protein). BIP mRNA

harbors upstream open reading frames (uORFs) in its 5'UTR and eIF2A functions as a cis-acting regulatory element required for UUG-initiated uORF translation during the ISR [20].

eIF2A homologs are found in a wide range of eukaryotic species, suggesting conserved physiological function. In *Saccharomyces cerevisiae*, eIF2A homolog is coded by the *YGR054W* non-essential gene [21]. This yeast eIF2A protein shares 28% identity and 58% similarity with the human eIF2A [21]. eIF2A specifically associates with the 40S and 80S ribosomal subunits but is not found in the polysome fraction. Furthermore, eIF2A physically and genetically interacts with the translation initiation factors eIF5B and eIF4E. Indeed, *eIF2A* deletion associated with either *eIF5B* deletion or with an *eIF4E-ts* mutation, strongly delayed cell growth compared to the wild-type strain [21, 22]. Moreover, eIF2A acts as a negative regulator of IRES-mediated translation of *URE2* mRNA, encoding a regulator of nitrogen metabolism [22, 23].

Both mammalian and yeast studies show that under standard conditions, eIF2 is the major initiation factor, while eIF2A appears to act in an alternative translation initiation pathway [24 for review]. However, considering the diverse roles reported for eIF2A, the function of this protein is currently still obscure.

In this study, we showed that eIF2A specifically bound mRNA encoding proteins required for cell wall biogenesis. *EIF2A* overexpression did not induce major changes in the transcriptome, but decreased the amount of several cell wall proteins, strongly suggesting that eIF2A controls expression of its mRNA target at the translational level. We also showed that eIF2A interacted with the RNA binding protein, Ssd1, independently of RNA and this interaction was required for eIF2A binding to some of its targets. Altogether, these results are consistent with a role of eIF2A as a protein interfering with translation of a very specific population of mRNAs coding for cell wall-related proteins.

## Materials and methods

### Yeast and *E. coli* strains, growth conditions

Yeast strains used in this study are listed in S1 Table. All *S. cerevisiae* strains were derived from BY4741 and grown at 30°C in YPGlu rich medium or in -URA or -HIS minimal medium. When necessary, media were supplemented with antibiotics used at the following concentrations: 0.5 mg/ml G418, 0.25 mg/ml hygromycin or 10 μg/ml doxycycline to repress gene expression under the control of the $P_{tet-off}$ promoter or with 100 μM Auxin/IAA (SERVA), to deplete Xrn1 protein fused to the degron.

The yeast strains were constructed by homologous recombination using PCR fragments to transform appropriate strains (S1 Table). The pAG32 plasmid was used to replace the KanMX6 cassette by the HphMX4 marker, conferring hygromycin resistance (S2 Table).

NEB 10-beta *E. coli* competent cells were used as the general cloning host and were grown at 37°C in LB medium supplemented with 50 μg/ml ampicillin.

To induce cell wall damage, CFW (Fluorescent Brightener 28 disodium salt solution, Sigma-Aldrich) was added at a final concentration of 100 μg/ml for -URA medium and either 150 or 250 μg/ml for YPGlu medium.

**DNA manipulation.** Plasmid DNA was extracted from *E. coli* using the NucleoSpin plasmid miniprep kit (Macherey-Nagel). *S. cerevisiae* chromosomal DNA was isolated as previously described [25]. PCRs were carried out from bacterial colonies with Q5 high-fidelity DNA polymerase (NEB) to amplify DNA fragments used for cloning or strain constructions. PCR products were purified using a PCR cleanup kit (Macherey-Nagel).

**Plasmid constructions.** Plasmids used in this work are listed in S2 Table. The *eIF2A* and *SSD1* coding sequences were amplified using the appropriate oligonucleotides (S3 Table) with genomic DNA from BY4741 strain as the template. The PCR fragments were digested by

BamH1/Not1 enzymes and then ligated into the expression vector pCM190 under the control of the $P_{tet-off}$ promoter, repressed by doxycycline.

***eIF2A* or *SSD1* overexpression.** A pool of transformants harboring pCM190: *eIF2A* or pCM190: *SSD1* vectors were grown at 30°C in -URA medium containing doxycycline up to an $OD_{600nm}$ 0.6. Cells were harvested, washed twice with -URA medium without doxycycline and diluted to an $OD_{600nm}$ 0.08 in -URA medium devoid of doxycycline. After 5 hours at 30°C with shaking, to allow *eIF2A* or *SSD1* overexpression, cells were harvested and the pellet was stored at -80°C.

**Western Blot analysis of Tos1, Ccw14, Sun4 and Cln1 TAP-tagged proteins.** C-terminal TAP tagged strains were washed, resuspended in SSC buffer (0.15M NaCl, 0.015M sodium citrate) and subjected to three cycles of vigourous vortexing (60 sec., at 4°C, 6m/sec, MagNA lyser, Roche) in the presence of acid-washed glass beads V/V (425–500 μm Sigma). Protein extracts were recovered by centrifugation (14000 rpm, 4°C, 10 min) and quantified by Bradford assay (Bio-Rad protein assay dye reagent). Then, 5 μg of proteins were denatured by SB2X (50mM Tris-HCl pH 6.8, 20% glycerol, 4% SDS, 100mM DTT and 0.05% bromo-phenol-blue), heated at 100°C for 3 min. and separated into a gradient polyacrylamide gel (NuPAGE 4–12% Bis-Tris Gel from Invitrogen) and transferred onto Nitrocellulose Membranes 0.45μm (Bio-Rad) using Trans-Blot turbo transfert system (Bio-Rad), according to the manufacturer's protocol. The membrane was incubated in blocking buffer (PBST with 5% milk) for 1 hour and then incubated with the appropriate antibody and dilution (S4 Table) for 1 hour at room temperature. Proteins were detected using the clarity ECL substrate (Bio-Rad) and Gel Doc XR system (Bio-Rad) according to the manufacturer's protocol.

**eIF2A-TAP affinity purification for mass spectrometry analysis and RNA sequencing.** The yeast strain expressing eIF2A-TAP protein was grown in YPGlu medium overnight at 30°C with shaking. We used a strain in which the catalytic site of Xrn1 was mutated (D206A) in order to globally prevent mRNAs from degradation. Upon reaching $OD_{600nm}$ 2.0, two liters of culture were centrifuged at 4°C, cell pellets were washed with cold water and subsequently stored at -80°C. Cells were then resuspended in 1mL of cell lysis buffer per gram of cells (20mM Hepes pH7.4, 10mM $MgCl_2$, 100mM KOAc) containing a protease-inhibiting reagent (Roche). The cell suspension was then added to 500 μL/mL acid-washed glass beads and vortexed three times for 40 sec at 4°C, 6m/sec (MP FastPrepTM, Fisher Scientific). The obtained cell lysate was clarified by centrifugation (14000 rpm, 4°C, 20 min), 0.5% Triton was added, and an aliquot of lysate was taken prior to eIF2A-TAP purification for analysis of total proteins (input). Next, 25 μL of covalently coupled IgG-Dynabeads® magnetic beads were resuspended in lysis buffer and added to the remaining supernatant and incubated for 2 hours at 4°C with gentle agitation. The beads were harvested and washed five times with washing buffer (20mM Hepes pH 7.4, 10mM $MgCl_2$, 100mM KOAc, 0.5% Triton) and once in lysis buffer. Proteins associated with eIF2A-TAP were eluted by incubation in elution buffer (2% SDS and 1X TE), combined with heat treatment at 65°C for 15min. and gentle agitation (300 rpm). For mass spectrometry analysis, SDS was removed from the supernatant using a HiPPR$^{TM}$ Detergent Removal Resin kit (Thermo Fisher Scientific) and proteins were precipitated by the Methanol/Chloroform method [26].

For Co-Immunoprecipitation, yeast strains expressing eIF2A-TAP or Ssd1-TAP and Ssd1-HA or eIF2A-HA (S1 Table) were used and eIF2A-TAP or Ssd1-TAP purification was done as described above, except that beads were washed 3 times and before elution of Ssd1 or eIF2A-associated complexes, samples were treated with 1 μL of micrococcal nuclease or nothing (NEB, $2.10^6$ U/mL) for 10 min. at 37°C in washing buffer supplemented with 1mM $CaCl_2$. RNA extraction was performed (see below RNA extraction and Northern Blot) on remaining samples to confirm its absence after RNAse treatment. Proteins were denatured, separated

into a polyacrylamide gel and detected with the appropriate antibody (S4 Table) using clarify ECL substrate (Bio-Rad), as described above (see Western Blot analysis).

For RIP-Seq experiment, upon reaching $OD_{600nm}$ 0.8, twelve liters of culture were centrifuged, and eIF2A-TAP purification was carried out as described above, except that 1μl of RNasin ribonuclease inhibitors (Promega) was added per ml of lysis and washing buffers. Cells were broken by vortexing two times for 90 sec. at 3000 rpm (MagNAlyser, Roche). The lysates were recovered by centrifugation (20 min., 14 000 rpm, 4°C) and incubated with magnetic beads for 1 hour at 4°C. The beads were harvested and washed as described above except that the last washing was done with buffer not containing $MgCl_2$ (20mM Hepes pH7.4, 100mM KOAc, 1μL/mL RNAsin). mRNAs associated with eIF2A-TAP were resuspended in elution buffer (2% SDS, 1X TE, 30mM EDTA), eluted after heat treatment at 65°C for 15 min. and extracted as described below (see RNA extraction and Northern Blot). An aliquot of lysate was taken prior to eIF2A-TAP purification for the analysis of total RNA (input).

**RNA extraction and Northern Blot analysis.** RNAs were purified using hot acid phenol/chloroform (Phenol: Chloroform: Isoamyl Alcohol 25:24:1, Interchim) [27] and then precipitated overnight with cold 1M ammonium acetate and ethanol at -20°C. Samples were centrifuged (20 min., 14 000 rpm, 4°C) and the pellets were washed with cold 70% EtOH before being resuspended in $H_2O$. RNAs were denatured by formazol (FO 121, MRC) and heat-treatment at 65°C for 5 min. and then separated on a 1.25% agarose gel in migration buffer 1X TBE (tris borate EDTA).

After transfer on Nylon membrane (BrightStar$^{TM}$ Plus, Invitrogen), RNAs were UV cross-linked at 0.120 Joules and revealed with strand specific DIG-labeled riboprobes. RNA probes were transcribed using the DIG RNA labelling Kit (SP6/T7, Roche) and as template, pre-annealed oligonucleotides or PCR products including T7 promoter (S3 Table). Probes were then purified using illustra microspin G-25 columns (GE Healthcare) and diluted in hybridization buffer (Ambion ULTRAhyb® ultrasensitive, Invitrogen). After denaturation at 85°C for 5 min., RNA probes were used to hybridize Nylon membrane overnight at 60°C or 65°C for probes generated from pre-annealed oligonucleotides or PCR products, respectively. After washing using Wash and Block buffer Set (Roche), DIG-labelled RNA probes were revealed with the Anti-DIG antibody (S4 Table) and clarity ECL substrate (Bio-Rad).

## Illumina RNA sequencing

Five micrograms of RNA were depleted for abundant ribosomal RNA using Ribominus transcriptome Isolation kit (Invitrogen) and the remaining RNAs were then sequenced by using TruSeq Stranded mRNA kit (Illumina). Briefly, mRNA samples were chemically fragmented and used as templates to be transcribed into first strand complementary DNA (cDNA) using reverse transcriptase and random primers. The second strand cDNA was then produced and after purification with AMPure beads, the 3' ends of the blunt fragments were adenylated and index adapter sequences were added by PCR amplification, generating a dual-indexed library. The resulting products were purified using AMPure beads and the concentration and quality of the libraries were checked by Qubit and Bioanalyzer (Agilent). Libraries were pooled at a final concentration of 2.1 pM, denatured and sequenced with a NextSeq500 sequencing system.

## RNA seq data analysis

After demultiplexing and removal of adapter sequences from Fastq files with Cutadapt [28], reads were mapped on *S. cerevisiae* S288C genome using RNA STAR [29]. Default parameters were used except for maximum intron size (1500), maximum gap between two mates (1500), minimum overhang for spliced alignments (25) and the annotated GTF file *Saccharomyces cerevisiae* (R64-1-1.104) from ENSEMBL was used for mapping. Indexed BAM files were

generated using Samtools_sort [30] and then read counts were obtained using featureCounts [31] and a GTF file from [32], was used as the gene annotation file. Default parameters were used except that both multi-mapping and multi-overlapping features were included. Differential analysis of RNA-Seq data from three independent biological replicates was performed by SAR-Tools, using DESeq2 software [33]. To select the most significantly enriched targets based on a single factor, we multiplied $\log_2$ Fold Change by $-\log_{10}$ ($p$-value) to generate a Vfactor, which depends both on the level of variation and the significance. We retained the mRNAs having a Vfactor >50. This filter led to a selection of 146 targets as the most enriched mRNAs by eIF2A.

## LC-MS acquisition

Briefly, after reduction and alkylation, protein samples were treated with Endoprotease Lys-C (Wako) and Trypsin (Trypsin Gold Mass Spec Grade; Promega). LC-MS/MS analysis of digested peptides was performed on an Orbitrap Q Exactive Plus mass spectrometer (Thermo Fisher Scientific, Bremen) coupled to an EASY-nLC 1200 (Thermo Fisher Scientific). Mass spectra were acquired in data-dependent acquisition mode with automatic switching between MS and MS/MS scans using a top-10 method.

## Protein database search

All RAW files were processed together in a single run by MaxQuant [34] version 2.0.3.0 with default parameters unless otherwise specified (http://www.maxquant.org). Database searches were performed with the built-in Andromedasearch engine against the reference yeast proteome (downloaded on 2021.10.09 from Uniprot, 6050 entries). Precursor mass tolerance was set to 6 ppm in the main search, and fragment mass tolerance was set to 20 ppm. Digestion enzyme specificity was set to trypsin with a maximum of two missed cleavages. A minimum peptide length of 7 residues was required for identification. Relative label-free quantification of proteins based on intensities was done using the MaxLFQ algorithm integrated into MaxQuant [35]. Proteins that shared same identified peptides were combined into a single protein group.

## Proteomic data analysis

To identify interactors, replicates of affinity-enriched bait samples were compared to a set of negative control samples (n≥3). Proteomics data analysis was performed in the Perseus environment (version 1.6.15) (https://maxquant.org/perseus/) [36]. "Proteingroups.txt" file from MaxQuant was loaded. Protein groups identified by a single "razor and unique peptide" were filtered out from the data set. Protein group LFQ intensities were log2 transformed. A minimum of valid values (60%) was required in at least one group. Missing values were assumed to be biased toward low abundance proteins that were below the MS detection limit. Imputation of these missing values was performed separately for each sample from a distribution with a width of 0.3 and downshift of 1.8. Student's $t$-test calculations, used in statistical tests of LFQ intensities, implemented in Perseus, showed that all data sets approximated normal distributions, with FDR = 0.01 (False Discovery Rate) [36]. Significant interactors were determined by a volcano plot-based strategy, combining $t$ test $p$-values with protein ratio information.

# Results

## eIF2A binds mRNAs encoding proteins required for the cell wall organization

While, the function of eIF2A is not yet fully understood, some data from the literature indicate that eIF2A could be a non-canonical translation initiation factor [24] and references therein].

To identify mRNAs associated with eIF2A, we performed an RNA-binding protein immu-noprecipitation followed by a sequencing (RIP-Seq) experiment. For this purpose, we affinity purified an eIF2A-TAP protein, which is functional, as it allowed the growth of the cells under stress conditions that affected the *eif2aΔ* strain, as shown in S1A Fig. Total and eIF2A-associ-ated mRNAs from exponential-phase culture were extracted, reverse transcribed into cDNA and sequenced. Enrichment of each mRNA in the immunoprecipitated fraction relative to total RNA was calculated from three independent biological replicates and normalized using DESeq2 [33] from three independent biological replicates. We found 146 mRNAs significantly enriched with eIF2A, according to the selection criteria that we applied (see Materials and Methods) (Fig 1, S5 Table and S1 Dataset). Gene ontology (GO) term enrichment analysis was done on the Saccharomyces Genome Database (SGD) site according to [37]. It revealed that 24% of the selected mRNAs encode proteins required for cell wall biogenesis with a *p*-value of 6.56 e-22. This high percentage revealed a strong enrichment for mRNAs involved in the cell wall biogenesis, while this class of mRNAs represented less than 3% of the total mRNA [38]. Among the most highly enriched mRNAs, we found *TOS1*, *CCW14*, which encode covalently bound cell wall proteins; *SUN4*, which encodes a glucanase localized in bud scars; *CTS1* and *SRL1*, which respectively encode an endochitinase and a mannoprotein exhibiting a tight asso-ciation with the cell wall (Fig 1, S5 Table and S1 Dataset). Other eIF2A-enriched mRNAs are related to the endoplasmic reticulum (11%), plasma membrane (7%), mitochondria (8%) or are involved in cell cycle regulation (6%) such as the G1/S cyclins *CLN1*, *CLN2 and CLN3* (Fig 1, S5 Table and S1 Dataset).

The specific association of cell wall biogenesis related mRNAs with eIF2A suggests that this protein could play a role in cell wall homeostasis. The yeast cell wall is composed of a network including beta-1,3-glucan and 1,6-glucan, chitin and mannoproteins [39]. We decided to address eIF2A involvement in the control of cell wall integrity by using calcofluor white (CFW), a drug which binds chitin and interferes with cell wall biogenesis [40]. Compared to the wild-type strain, growth of the *eif2aΔ* mutant was not significantly affected in YPGlu rich medium (Fig 2A) but was delayed when cells were cultivated in the presence of CFW (Fig 2B). This slow-growth phenotype was fully restored by *eIF2A* episomal expression in the *eif2aΔ* mutant strain (S1B Fig). Note that to limit the *eIF2A* induction and consequently, to avoid the toxicity, the cells were plated on YPGlu instead on -URA. In contrast, *eIF2A* overexpression from pCM190:*eIF2A* vector, severely impaired wild-type strain growth under standard condi-tions (Fig 2C), but CFW addition did not increase the cell growth defect observed when *eIF2A* was overexpressed, suggesting that *eIF2A* overexpression is epistatic over the effect of CFW (Fig 2D).

Together, our results are consistent with eIF2A association with cell wall-related mRNAs and strengthen the hypothesis that eIF2A could be involved in the expression of transcripts required for cell wall biogenesis.

To validate RIP-Seq results, we investigated four identified eIF2A mRNA targets (*TOS1*, *CCW14*, *SUN4*, *CLN1*) by Western Blot analysis, to further explore the behavior of the corre-sponding TAP-tagged proteins upon *eIF2A* overexpression. Interestingly, *eIF2A* overexpres-sion reproducibly decreased the amount of Tos1, Ccw14 and Sun4 proteins by approximatively 2-fold and by 5-fold in the case of Cln1, while as a control, G6PDH protein level did not change (Fig 3A and 3B).

In agreement with the RIP-Seq data (Fig 1, S5 Table and S1 Dataset), these results confirm that eIF2A protein is involved in the regulation of *TOS1*, *CCW14*, *SUN4* and *CLN1* gene expression (Fig 3).

To further explore whether the protein level changes observed upon *eIF2A* overexpression conditions occurred at the transcriptional or translational level (Fig 3), we performed a

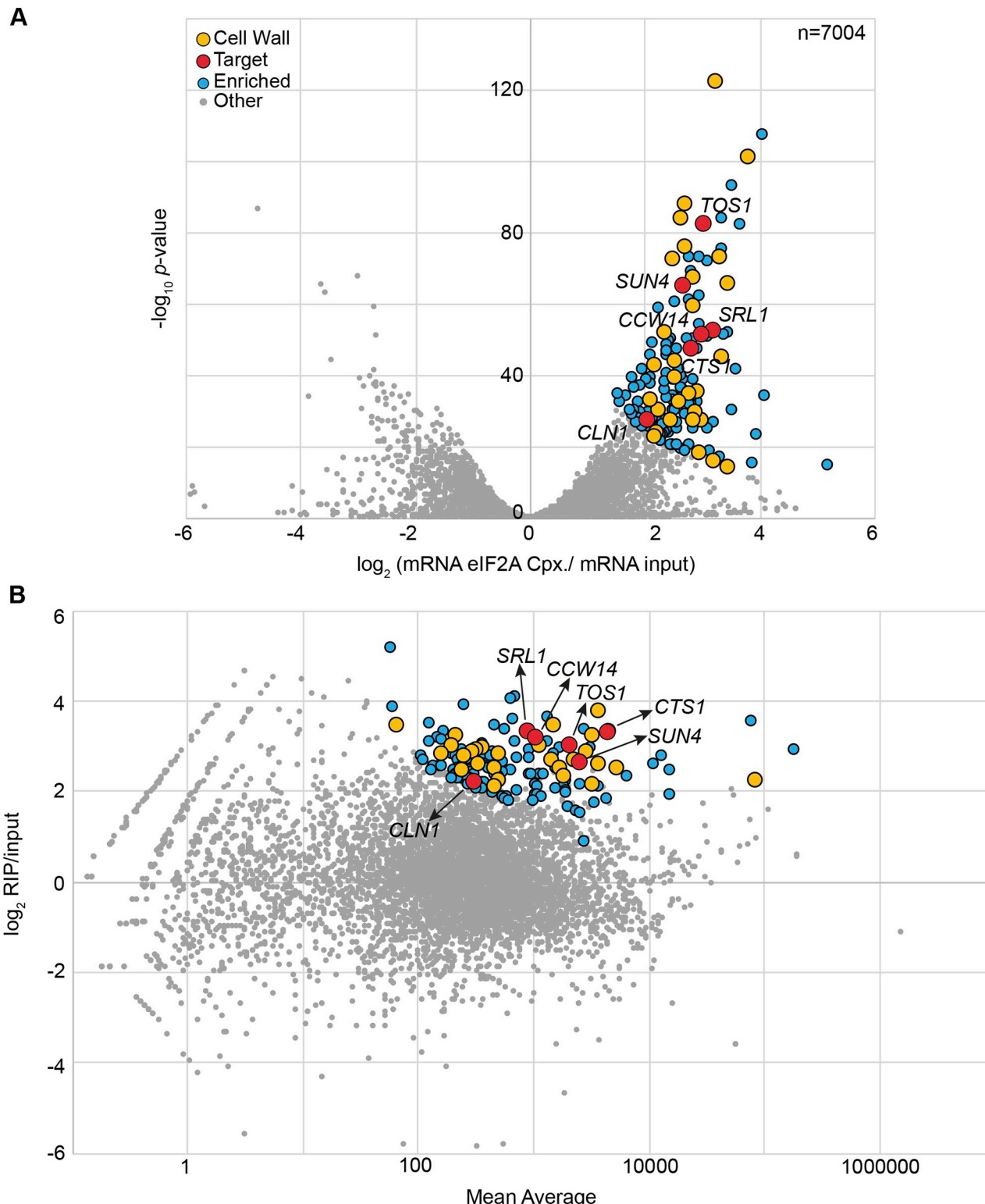

**Fig 1. mRNAs related to cell wall biogenesis are enriched in the complex associated to eIF2A.** (A) Volcano plot of RNA sequencing results. The x-axis displays the $\log_2$ fold change between the average number of reads in the eIF2A-associated fraction relative to the total RNA fraction, while the y-axis displays the $-\log_{10}$ of the associated $p$-value. Significantly enriched transcripts (as described in Material and Methods) are displayed in blue, with a yellow overlay for cell wall-related mRNAs. Red dots indicate candidates that were used for functional analyzes (see below). (B) MA plot. The x-axis displays the average number of reads between the two conditions ($\log_{10}$ scale), while the y-axis represents the $\log_2$ fold change between the average number of reads in the eIF2A-associated fraction relative to the total RNA. Colored dots correspond to the same categories as those mentioned in (A).

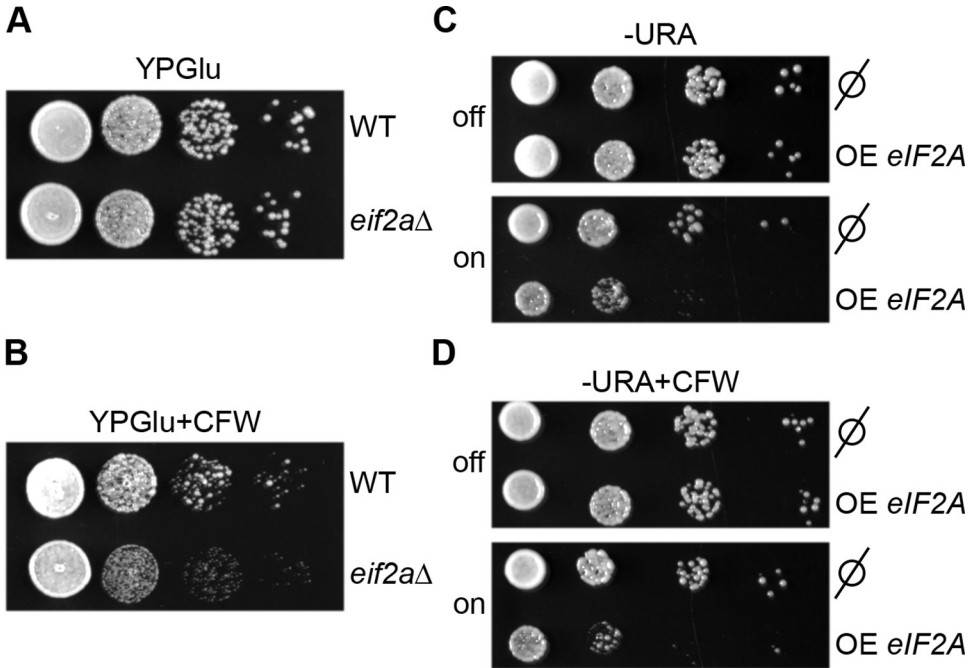

**Fig 2. eIF2A absence is detrimental for *S. cerevisiae* when the cell wall is affected, and its overexpression is detrimental under standard conditions.** (A and B) Wild-type strain and cells deleted for *eIF2A* gene were spotted in $10^{-1}$ dilution series on rich medium plates without CFW (A) or with (B) and incubated at 30˚C for 40 hours. (C and D) Wild-type strains harboring either empty pCM190 (ø) or pCM190: *eIF2A* vectors, allowing *eIF2A* overexpression (OE *eIF2A*) (on) or not (off), were serially diluted and spotted on (-URA) minimal medium supplemented (D) or not (C) with CFW. The precultures were done in the presence of doxycycline (Dox) to prevent the expression of *eIF2A* which is under the control of the $P_{tetoff}$ and spotted on a -URA without Dox. Plates were incubated at 30˚C for 40 hours.

genome-wide RNA sequencing (RNA-Seq) analysis. For that purpose, we compared the transcriptome of strains harboring either pCM190: *eIF2A* or the pCM190 empty vector and showed that no major changes in the general mRNAs levels were detected after 5 hours of *eIF2A* overexpression (Fig 4A and S2 Dataset). Similarly, compared to the wild-type strain, the absence of eIF2A did not substantially disturb the transcriptome (Fig 4B and S2 Dataset), suggesting that eIF2A does not control the expression of its mRNA targets at a transcriptional level.

Together, our results combined with the fact that eIF2A associates specifically with 40S and 80S ribosomal subunits [21], strongly suggest that eIF2A acts as a negative translational regulator of a class of mRNAs involved in the cell wall organization and biogenesis.

### The RNA binding protein Ssd1 is highly enriched by eIF2A independently of RNA

To gain further insight into the mechanistic role of eIF2A, we performed an *in vivo* affinity purification of proteins using eIF2A-TAP as bait. After purification on IgG coupled magnetic beads, the eIF2A-associated complex was analyzed on a silver-stained polyacrylamide gel (Fig 5A) and eIF2A interaction partners were identified and quantified by mass spectrometry (S3 Dataset). A total number of 1509 proteins were identified in either input or purified samples but only 1037 were quantified in both. The enrichment level for each protein is indicated in the S3 Dataset. A volcano plot (Fig 5B) represents analysis of the label-free quantitative (LFQ) MS data. This analysis highlighted that three groups of proteins were significantly enriched by

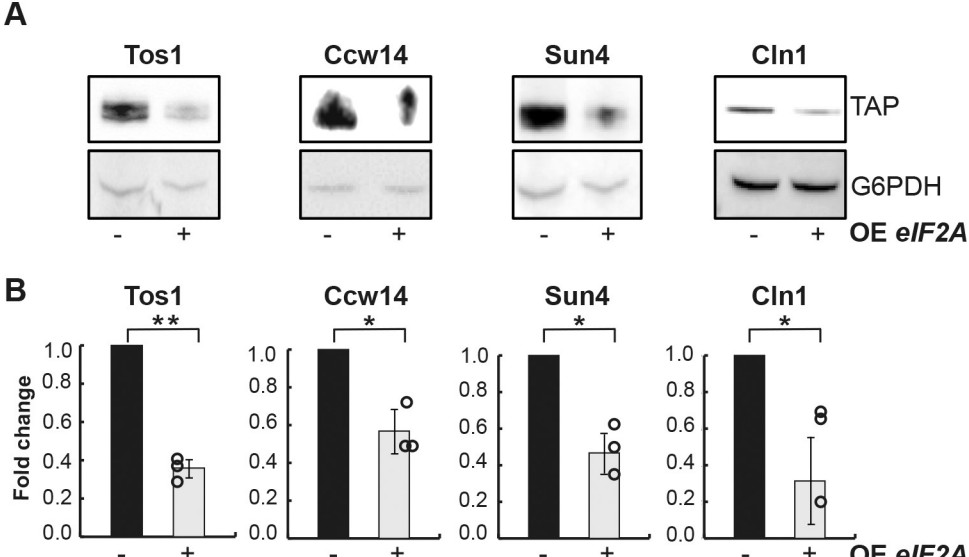

**Fig 3. eIF2A overexpression decreases the cell wall protein levels.** (A) Cells harboring pCM190 or pCM190: *eIF2A* plasmids and producing Tos1, Ccw14, Sun4 or Cln1 TAP-tagged proteins were grown in -URA medium and harvested 5 hours upon *eIF2A* overexpression (+) or not (-). Protein extracts were separated on a denaturing polyacrylamide gel and TAP-tagged proteins were revealed by Western Blot with PAP antibodies, as described in Materials and Methods. G6PDH was used as a loading control. (B) Quantification of Western Blot analyzes was performed using ImageJ and was based on the expression levels of target proteins relative to G6PDH reference protein. Fold change indicated on y-axis was defined as the ratio between relative abundance of target proteins in cells harboring pCM190: *eIF2A* (+) and pCM190 plasmids (-). Error bars indicate the standard deviations of averages for at least three independent experiments. Statistical analysis was performed by using a *t*-test, with the following obtained *p*-values: $p = 0.00182$; $p = 0.01818$; $p = 0.01077$; $p = 0.02992$ for Tos1, Ccw14, Sun4 and Cln1 respectively. Asterisks indicate statistical significances (*: *p*-value $\leq 0.05$, **: *p*-value $\leq 0.01$). The dots correspond to the value obtained for each individual replicate.

eIF2A. The most enriched proteins are related to mRNA degradation pathways, such as the 5'-3' exonuclease XrnI [41]; Ska1, a SKI complex-associated protein involved in degradation of mRNAs containing long 3' UTR devoid of ribosomes [42] (Fig 5B). We also found the decapping mRNA complex (Dcp1, Dcp2, Edc3 and Pby1) [43] and deadenylation-dependent mRNA decapping enhancers including the Lsm1-7 complex and Pat1 [44, 45] (see below) (Fig 5B). As expected, a second group of eIF2A-enriched proteins (RPL/RPS) is related to the small 40S and large 60S ribosomal subunits. Interestingly, we observed a robust enrichment of Ssd1 (Fig 5B). Ssd1 is a RNA-binding protein (RBP) and previous studies have reported that Ssd1 binds to about a hundred mRNA coding for proteins involved in cell wall biogenesis [46–49]. Ssd1 directly recognizes a consensus motif usually located in the 5' UTR of these mRNA targets [46].

We confirmed the eIF2A-Ssd1 interaction by Co-Immunoprecipitation using Ssd1-TAP or eIF2A-TAP as bait in a strain expressing eIF2A-HA or Ssd1-HA fusion proteins, respectively. We first showed the ability of tagged proteins to complement the CFW-sensitive phenotype of the *eif2aΔ* or *ssd1Δ* mutants (S1A Fig).

We next carried out purification of Ssd1-TAP or eIF2A-TAP interaction partners and verified by Western Blot that Ssd1 interacts with eIF2A (Fig 5C and S2A Fig). To see whether this interaction requires the presence of RNA, we performed a nuclease treatment before elution of Ssd1- or eIF2A-associated complex and showed that the eIF2A-Ssd1 interaction was RNA-independent (Fig 5C and S2A Fig). To ensure that RNA was properly digested, we extracted RNA from immunoprecipitated fractions and verified the absence of RNA in samples treated

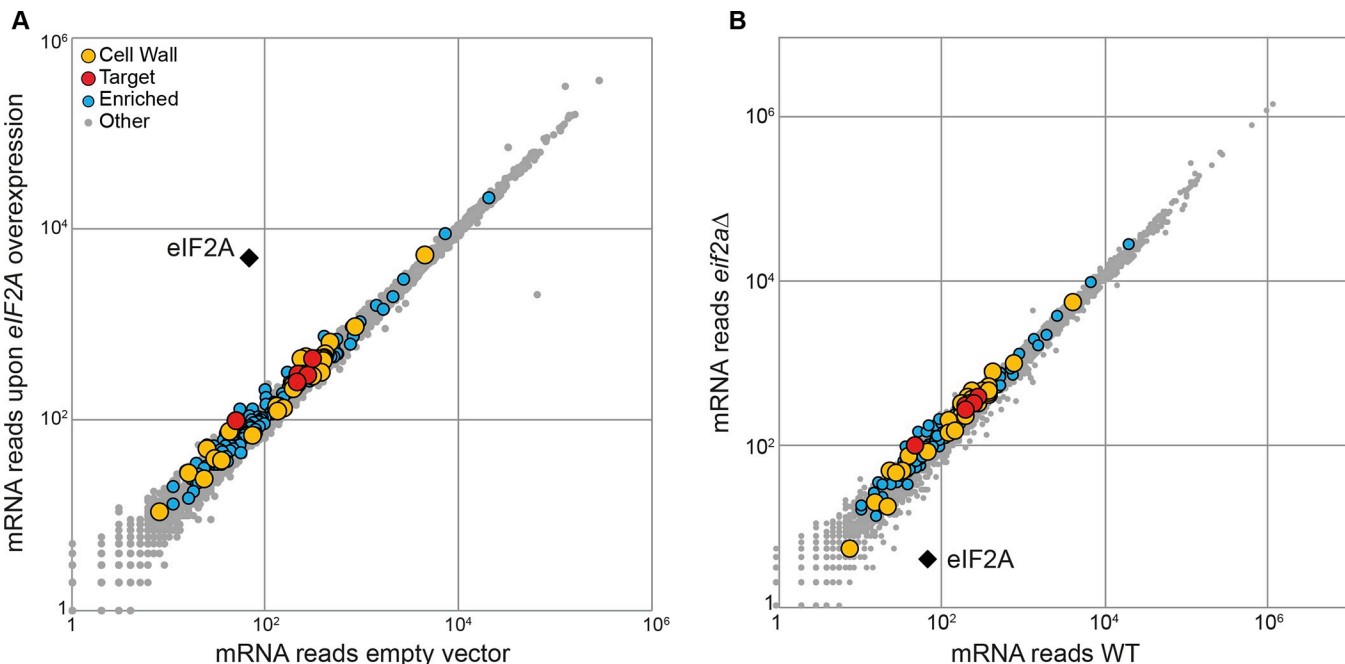

**Fig 4. Overexpression or deletion of *eIF2A* has no effect on the transcriptome compared to the wild-type strain.** (A) Scatter plot of RNA-Seq data comparing mRNA abundance between transformed strains with pCM190-*eIF2A* or pCM190 plasmids. Cells were grown in–URA medium to exponential-growth phase and harvested after 5 hours of *eIF2A* overexpression or not. The x-axis and the y-axis represent the mRNA levels of strains harboring pCM190 or pCM190: *eIF2A* respectively. (B) Scatter plot comparing transcriptomes of WT and *eif2aΔ* strains transformed with pCM190 plasmid and grown on -URA medium to exponential-growth phase. The color dots code is the same as in Fig 1.

with micrococcal nuclease compared to untreated samples (S2B Fig). In contrast to Co-Immunoprecipitation of eIF2A-TAP-associated complex, which confirmed that eIF2A-TAP interacted with Ssd1-HA protein and XrnI (S2A Fig), Ssd1-TAP did not seem to interact with the 5'-3' exonuclease XrnI (Fig 5C).

### eIF2A requires the presence of Ssd1 to associate with several of its mRNA targets and to regulate the expression of *SUN4*

RNA binding proteins (RBPs) play an important role in post-transcriptional control of fungal cell wall biogenesis and several studies characterized many RBPs to identify their RNA targets. In a systematic RIP-Seq study, at least four of a set of six cell wall-related RBPs (Ssd1, Khd1/Hek2, Pub1, Mrn1, Scp160 and Nab6) [50 for review] enriched a common subset of 78 mRNAs, suggesting that RBPs act together to regulate cell wall organization.

We observed that *eIF2A* deletion did not increase the sensitivity of the *ssd1Δ* mutant to CFW (S3 Fig), This epistatic behavior of *ssd1Δ* over *eif2aΔ* suggests that Ssd1 and eIF2A act in the same regulatory pathway in response to cell wall damage.

Comparison of Ssd1-associated mRNAs identified by CRAC analysis [46] with previous RIP and transcriptome data [47–49] showed that 11 mRNAs were systematically bound by Ssd1 (Table 1). Among these 11 mRNAs, 8 were also highly enriched by eIF2A, including *CTS1*, *SUN4* and *SRL1* (Fig 1, Table 1, S5 Table and S1 Dataset). As mentioned above, Ssd1 directly binds its mRNA targets [46], while currently none of our results demonstrated that this is the case for eIF2A. Given that Ssd1 and eIF2A seem to share the regulation of several targets, we tested the hypothesis that eIF2A binds some of its mRNA targets through Ssd1.

For that purpose, we performed a RIP experiment of eIF2A in the presence or absence of Ssd1 and evaluated the associated mRNA by Northern Blot analysis. In agreement with our

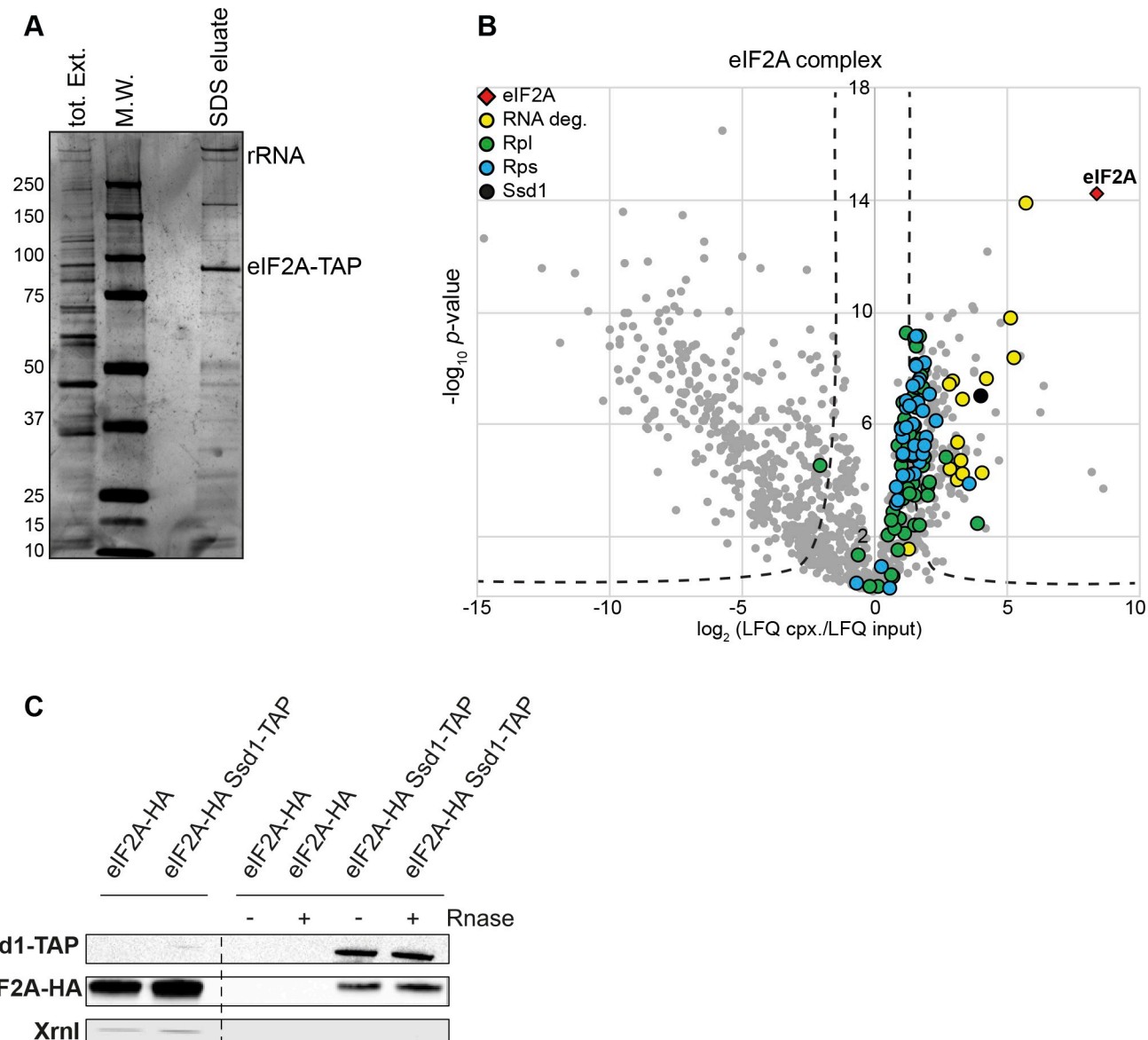

**Fig 5. eIF2A interacts with the RNA binding protein Ssd1.** (A) Affinity purification using eIF2A-TAP as bait. Total proteins (tot. Ext.) and eIF2A-associated complex (SDS eluate) were separated on a polyacrylamide gel and visualized by silver staining. MW: Molecular Weight marker. (B) Volcano plot showed proteins enriched by eIF2A-TAP identified by mass spectrometry (LC-MS/MS). The x-axis represents the $\log_2$ fold change of each protein in the eIF2A-TAP enrichment from the lysate. The y-axis shows $\log_{10}$ p-value calculated using a Student's t-test. Proteins above the curved lines on the right part of the plot are significantly enriched by eIF2A (visualized by red diamond) purification. Proteins linked to RNA degradation are indicated by yellow dots. Proteins of the small 40S or large 60S ribosomal subunits are visualized by blue or green dots, respectively. Results are from six independent experiments. (C) The interaction between eIF2A and Ssd1 was confirmed by Co-Immunoprecipitation of eIF2A-HA with Ssd1-TAP. Cells expressing eIF2A-HA and Ssd1-TAP proteins were cultivated to exponential-growth phase and Ssd1-TAP and its interaction partners were purified as described in Materials and Methods. The Ssd1-associated complex was eluted after a nuclease treatment (+) or not (-) using micrococcal nuclease. A strain lacking the TAP-tag fused to the Ssd1 protein was used as a control. Total (input) as well as purified proteins were separated on a polyacrylamide gel and TAP- or HA-tagged proteins were revealed by Western Blot with PAP or HA-tag antibodies, respectively.

RIP-Seq data (Fig 1, S5 Table and S1 Dataset), we confirmed that *SUN4*, *CTS1*, *SRL1* mRNAs were strongly enriched by eIF2A-TAP in the wild-type strain, while *SSD1* deletion dramatically decreased eIF2A association with these mRNAs (Fig 6). However, we found that the eIF2A binding to these mRNAs targets was not always Ssd1-dependent, for example, *CCW14* mRNA

**Table 1. eIF2A shares common targets with Ssd1.** Ssd1 mRNA targets systematically identified by RIP [47–49] and CRAC [46] experiments. The functions were extracted from SGD https://www.yeastgenome.org/.

| Ssd1 targets | eIF2a targets | Function |
|---|---|---|
| *TOS1* | + | Covalently-bound cell wall protein of unknown function |
| *LRE1* | | Protein involved in control of CW structure and stress response |
| *SCW4* | + | CW protein with similarity to glucanases |
| *DSE2* | + | Daughter cell-specific secreted protein, similarity to glucanases |
| *SIM1* | + | Protein of the SUN family (Sim1p, Uth1p, Nca3p, Sun4p) |
| *HSL1* | | Septin-binding kinase that localizes to the bud neck septin ring |
| *UTH1* | + | Mitochondrial inner membrane protein implicated in CW biogenesis |
| *MMR1* | | Mitochondrial Myo2p Receptor-related |
| *CTS1* | + | Endochitinase |
| *SUN4* | + | CW protein related to glucanases localized in birth scars |
| *SRL1* | + | Mannoprotein that exhibits a tight association with the CW |

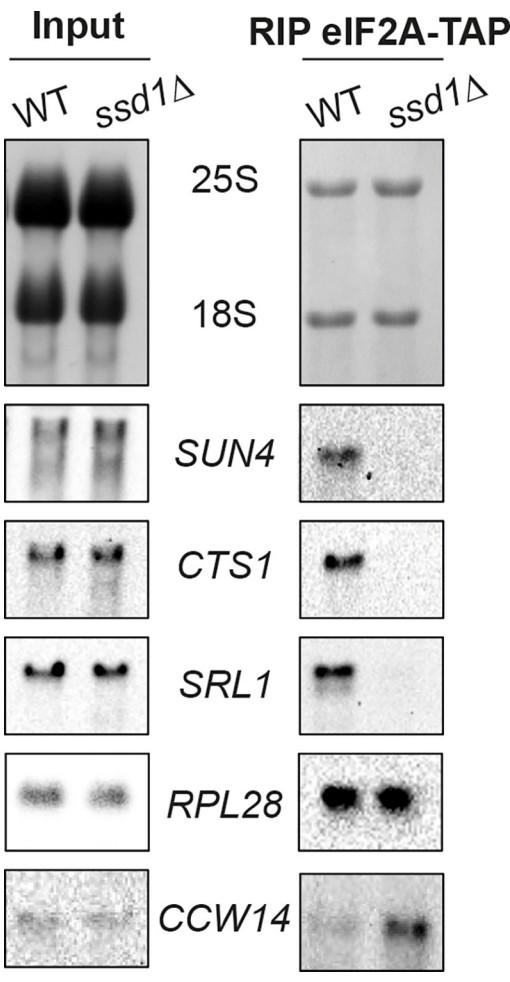

**Fig 6. The *SUN4, CTS1, SRL1* mRNAs are enriched by eIF2A-TAP only in the presence of Ssd1.** Wild-type and the *ssd1Δ* mutant strains expressing eIF2A-TAP protein were cultivated to exponential-growth phase. RIP experiment was performed as described in Materials and Methods. 8 μg of total RNA (input) and 1 μg of immunoprecipitated RNA (RIP eIF2A-TAP) were respectively separated by an agarose gel. *SUN4, CTS1, SRL1* and *RPL28* mRNAs were revealed by Northern Blot with appropriate DIG-labeled probes and anti-DIG antibody.

was still enriched by eIF2A even in the absence of Ssd1 (Fig 6). As a control, we observed that the *RPL28* mRNA encoding the 60S ribosomal protein L28, was not differentially enriched by eIF2A in the presence or absence of Ssd1 (Fig 6).

Finally, due to the impact of Ssd1 on eIF2A binding to target mRNAs, a 2-fold decrease in the Sun4 protein amount upon *eIF2A* overexpression was abolished in the absence of Ssd1 (Fig 7). Taken together, our results show that the presence of Ssd1 is required for the binding of eIF2A to *SUN4* mRNA and consequently, is needed to control its translation.

### eIF2A physically and genetically interacts with the 5'-3' exonuclease Xrn1

Surprisingly, in addition to Ssd1, we identified by affinity purification, a strong interaction between eIF2A and several players in the 5' to 3' mRNA degradation pathway, including the exoribonuclease Xrn1 [41], the decapping complex (Dcp1, Dcp2, Edc3 and Pby1) [43] and the cytoplasmic Lsm complex (Lsm1-7 and Pat1) [44, 45]. The label-free quantitative analysis of the eIF2A partners identified by mass spectrometry revealed that the most enriched factor was Xrn1 (Fig 5B). This result confirmed the interaction previously identified by Co-Immunoprecipitation using eIF2A-HA as bait [51]. Moreover, a Co-Immunoprecipitation experiment using eIF2A-TAP as bait confirmed the interaction between eIF2A and Xrn1 (S2 Fig). Notably, a Co-Immunoprecipitation experiment using Ssd1-TAP as bait did not purify Xrn1, whereas eIF2A-HA was highly enriched (Fig 5C). These results suggest that eIF2A, but not Ssd1, forms a sub-complex with the 5' to 3' mRNA degradation machinery.

To investigate a potential functional link between Xrn1 and eIF2A, in addition to the physical link, we looked at the effect of combining mutants affecting Xrn1 and eIF2A on viability. Since *XRN1* deletion affected the cell viability under standard rich culture conditions (Fig 8), we first achieved conditional depletion of Xrn1, using an auxin-inducible degron system (AID) in which rapid degradation of the protein is induced in the presence of auxin (IAA). We combined *eIF2A* deletion with a Xrn1-degron fusion to compare the cell viability when Xrn1 was depleted in the presence or absence of eIF2A. Under standard growth conditions, *eIF2A* deletion increased the growth defects of the *xrn1*-deg mutant in the presence of auxin (Fig 8). Interestingly, when we combined the double mutation with the presence of CFW, cell growth was dramatically affected (Fig 8 and S4 Fig). Thus, the synthetic slow-growth phenotype between Xrn1 depletion and the *eif2aΔ* mutation was amplified when the cell wall biogenesis was affected. We noted that Xrn1 depletion in the presence of CFW affected cell growth, highlighting a potential role of Xrn1 in the cell wall biogenesis or maintenance.

## Discussion

In eukaryotes, translation initiation is a highly regulated process, which involves many factors. Most mRNAs translation is initiated by eIF2-mediated binding of the initiator Met-tRNA to the 40S ribosomal subunit. However, an additional factor, eIF2A has been described to ensure persistent translation initiation of a subset of cellular and viral mRNAs under stressful conditions [24].

The data presented here revealed the negative role of eIF2A on the synthesis of proteins related to the cell wall biogenesis in *S. cerevisiae*. A large fraction of the most enriched mRNAs, in the eIF2A associated complex, belongs to the cell wall pathway (Fig 1). eIF2A was important for the cell fate when the cell wall integrity was disturbed (Fig 2). *eIF2A* overexpression led to a growth defect correlated with a decrease of several cell wall-related proteins encoded by the mRNA targets, notably *TOS1*, *CCW14* and *SUN4* (Fig 3), while no major changes were found in the transcriptome compared to the wild-type strain (Fig 4).

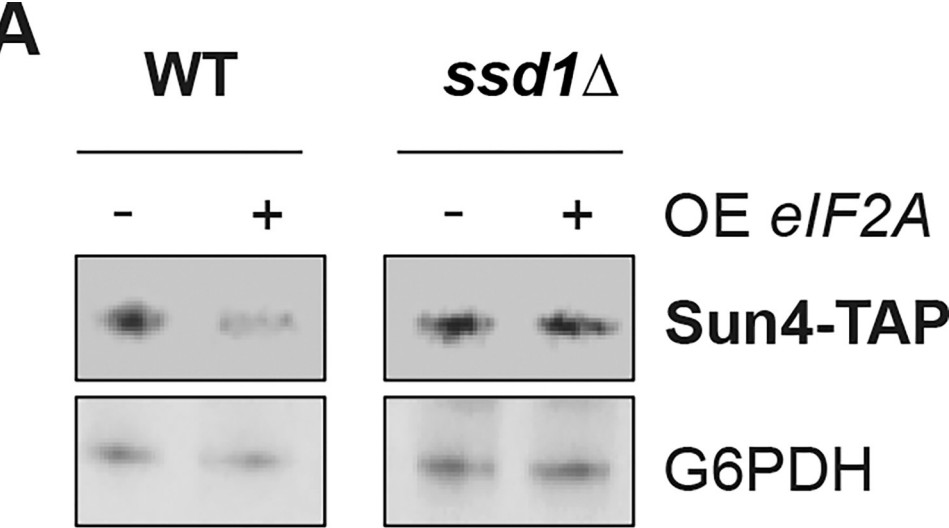

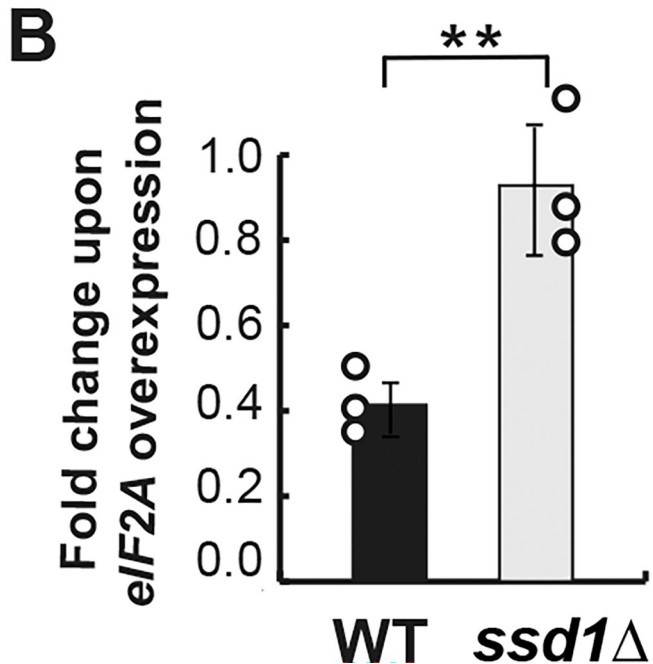

**Fig 7. Decrease of Sun4 protein level upon *eIF2A* overexpression requires the presence of Ssd1.** (A) Wild-type and the *ssd1Δ* mutant cells harboring pCM190 or pCM190: *eIF2A* plasmids and expressing Sun4-TAP protein were grown in -URA medium and harvested after 5 hours of *eIF2A* overexpression (+) or not (-). Protein extracts were separated on a polyacrylamide gel and Sun4-TAP was revealed by Western Blot using PAP antibodies. G6PDH was used as a loading control. (B) Quantification of Western Blot analyzes was performed as described in Fig 3B. Error bars indicate the standard deviations of averages from at least three independent experiments. Statistical analysis was performed using a *t*-test, *p* = 0.0054. The dots correspond to the value obtained for each individual replicate. Asterisks indicate statistical significances (*: *p*-value ≤0.05, **: *p*-value ≤ 0.01).

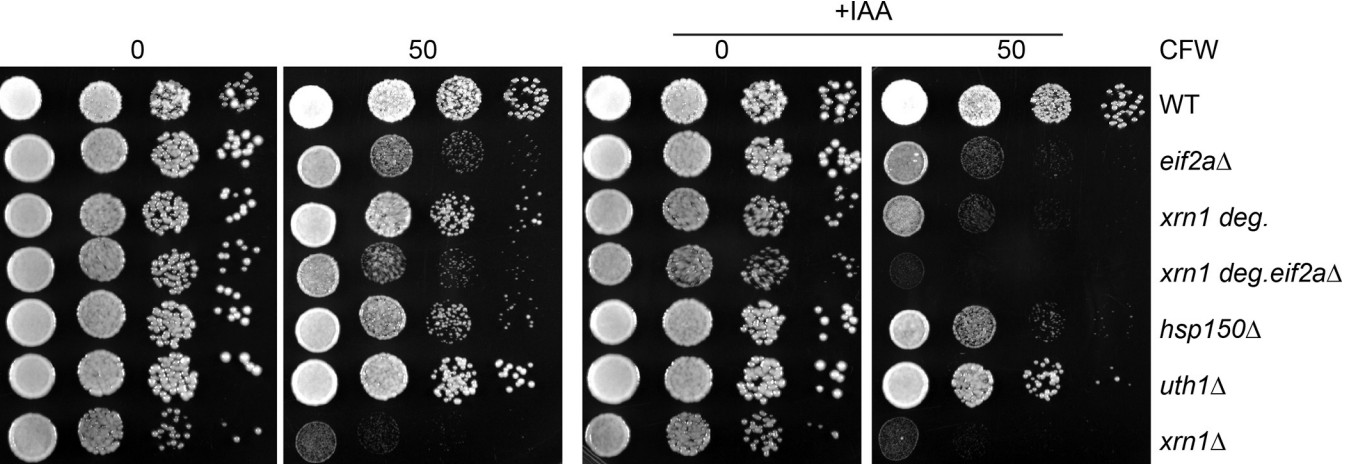

**Fig 8. Xrn1 depletion and eIF2A deletion are synthetic lethal.** Wild-type and mutant strains were serially diluted and spotted on YPGlu rich medium supplemented or not with CFW in the presence or not of IAA (100 μM Auxin) to deplete the cells of Xrn1. *uth1Δ* and *hsp150Δ* mutants were used as a control.

Taken together, our results combined with the fact that eIF2A-HA is associated with the 40S ribosomal subunit [22], strongly suggest that eIF2A acts as a negative translational regulator. Although eIF2A physically and genetically interacts with translation initiation factors eIF5B and eIF4E [21, 22], polysome profile analysis reported that eIF2A surprisingly associates with the 80S ribosomal subunits [21], which is unusual for initiation factors, commonly released from the 40S prior to 60S ribosomal subunit assembly. It has been previously proposed that eIF2A is slowly released from the initiation complex and participates in a late stage of translation initiation or during translation of the first amino acids, blocking the subsequent elongation step [21]. The exact function of eIF2A in translation regulation remains to be elucidated. The mammalian eIF2A mediates translation initiation of a subset of cellular and viral mRNAs under stress conditions, either through its recruitment to IRES [13, 15] or by initiating translation from non-AUG start codons or uORFs. It has been already reported in yeast, that eIF2A negatively regulates IRES-mediated translation of *URE2* mRNA [22, 23], and it would be of interest to investigate whether IRES or alternative start codons are found among the eIF2A mRNA targets.

The budding *S. cerevisiae* yeast cells are surrounded by a cell wall, which provides protection against environmental stress and maintains the shape and the rigidity of the cell. It is composed of crosslinked molecules, comprising β-1,3 glucans, β-1,6 glucans, mannoproteins and chitin [38 for review, 52]. Cell wall homeostasis is a dynamic process involving hundreds of proteins whose expression must be highly and quickly regulated by many RBPs, in response to environmental changes or depending on the cell cycle state [50 for review]. Ssd1 is one of the most studied RBPs linked to the cell wall. This protein directly binds to 5'UTR of many cell wall-related mRNAs to mediate translational repression of bound transcripts [46–49, 53]. Two other RBP, Nab6 and Mrn1, bind the 3'UTRs of mRNAs coding for cell wall proteins, but have antagonistic functions, supporting a model in which both proteins compete for RNA binding [53].

In contrast, in this study, biochemical characterization of the eIF2A-associated complex revealed that eIF2A robustly interacts with Ssd1 (Fig 5 and S2 Fig). We also showed that the double *ssd1Δ eif2aΔ* mutant displays the same CFW-sensitive phenotype as the *ssd1Δ* mutant (S3 Fig) and we found that Ssd1 and eIF2A share some mRNA targets (Table 1). Taken together, our results suggest that Ssd1 and eIF2A act in the same regulatory pathway in response to cell wall damage. Even though Ssd1 and eIF2A have overlapping mRNA targets

(Table 1), *eIF2A* or *SSD1* overexpression in the *ssd1Δ* or *eif2aΔ* mutants, respectively, did not restore, even partially, resistance to CFW (S1B Fig), suggesting that Ssd1 and eIF2A have distinct functions in the regulatory pathway controlling synthesis of cell wall-related proteins. Currently, none of our results demonstrate that eIF2A directly binds its mRNA targets and we observed that eIF2A-Ssd1 interaction is RNA-independent (Fig 5 and S2 Fig). This is consistent with the predicted structure of eIF2A, which harbors a WD-repeat ß-propeller fold in the N-terminal part but no RRM [24]. We also showed that Ssd1 is required for eIF2A binding to *CTS1*, *SUN4* and *SRL1* mRNAs (Fig 6) and consequently for the regulation of their expression, for example, *SUN4* (Fig 7).

More than 500 RBPs have been identified in *S. cerevisiae* and at least seven are related to cell wall synthesis with substantial overlap of their targets [47 for review, 54]. The multiplicity of the RBPs could exert synergistic effects on mRNA stability, localization or translation efficiency. We found that all eIF2A targets are not systematically shared by Ssd1 and some eIF2A and Ssd1 targets, for example, *CCW14* mRNA, is still enriched by eIF2A in the *ssd1Δ* mutant. This observation leads us to hypothesize that eIF2A might regulate some of its targets though the binding with other RBPs. The recruitment of several RBPs might lead to combinatorial effects that could allow a fine-tuned regulation of gene expression.

eIF2A also targets other mRNAs, including mRNAs involved in the cell cycle process, such as *CLN1*, *CLN2* and *CLN3* mRNAs. Since the cell wall biogenesis must be coordinated with the cell growth, it is not surprising that eIF2A can directly regulate the expression of cell cycle actors. A significant number of eIF2A mRNA targets are related to the mitochondria, plasma membrane and endoplasmic reticulum. Among those, Bgl2, Gas5 [55] or the SUN family genes, such as Sun4 and Uth1, have a cell wall and mitochondrial localization [56]. Furthermore, recently, Barbara Koch and Ana Traven proposed a model in which cell wall, plasma membrane, endoplasmic reticulum and mitochondria could be interconnected to respond to signal transduction during stress [57]. Our results suggest that eIF2A may be involved in the translation of a set of genes composed of mRNAs coding for macromolecular structure, which could be co-regulated in space and in response to environmental signals.

Finally, we highlighted that eIF2A physically interacts with several actors of mRNA degradation pathways, including the decapping complex and the cytoplasmic Lsm complex and we found that the most enriched eIF2A interactor is the 5'-3' exonuclease Xrn1 (Fig 5B). We also observed that eIF2A functionally interacts with Xrn1 and this interaction is even more prominent in response to cell wall damage induced by the presence of CFW (Fig 8 and S4 Fig). In contrast, Co-Immunoprecipitation experiments using Ssd1-TAP as bait, showed that Xrn1 is not enriched by Ssd1 (Fig 5C). The importance of post-transcriptional regulation by RBPs has been extensively characterized to ensure cell wall homeostasis [50 for review]. Recently, RNA exosome activity was shown to be necessary for maintaining cell wall stability [49, 58, 59] and interestingly, our results here indicate that additional control by the 5'-3' mRNA degradation machinery may exist. Further work is required to better understand the molecular function combining Xrn1 and eIF2A.

Here, we revealed the role of eIF2A as a new actor involved in the maintenance of cell wall homeostasis. The identification of post-transcriptional regulation mechanisms, controlling cell wall biogenesis in fungal species, can serve to better understand human fungal pathogens, enabling the design of novel antifungal therapeutic strategies.

## Supporting information

**S1 Fig. Functional and complementation assays for mutant strains.** (A) Wild-type, deletion mutants and strains expressing tagged proteins were plated in $10^{-1}$ dilution series on rich

medium with or without CFW and incubated at 30°C for 48 hours. (B) The CFW-sensitive phenotype of the *eif2aΔ* or *ssd1Δ* mutants were fully restored when *eIF2A or Ssd1* was trans-expressed in the *eif2aΔ* or *ssd1Δ* mutants, respectively. Wild-type strain and the *eif2aΔ* or *ssd1Δ* mutants harboring either empty pCM190 (ø), pCM190: *eIF2A* (OE *eIF2A*) or pCM190: *SSD1* (OE *SSD1*) vectors, were serially diluted and spotted on YPGlu rich medium supplemented with doxycycline and CFW (+) or not (-).
(TIF)

**S2 Fig. Interaction of eIF2A with Ssd1 was confirmed by Co-Immunoprecipitation.** (A) Cells expressing eIF2A-TAP and Ssd1-HA proteins were cultivated until exponential-growth phase and eIF2A-TAP and its interaction partners were purified. eIF2A-associated complex was eluted after a nuclease treatment (+) or not (-) using micrococcal nuclease. A strain lacking the TAP-tag fused to the eIF2A protein was used as a control. Total (input) as well as purified proteins were separated on a polyacrylamide gel and TAP-, HA-tagged and XrnI proteins were revealed by Western Blot with PAP, anti-HA or anti-XrnI antibodies, respectively. (B) RNA was extracted from Ssd1- or eIF2A-associated complexes and separated by agarose gel electrophoresis. As a control, 1 μg of RNA from the wild-type strain was loaded to visualize 25S and 18S ribosomal RNA.
(TIF)

**S3 Fig. The *ssd1Δ eif2aΔ* double mutant exhibits the same CFW-sensitive phenotype as the single *ssd1Δ* mutant.** Wild-type, *ssd1Δ*, *eif2aΔ* strains and the *ssd1Δ eif2aΔ* double mutant were serially diluted and spotted on YPGlu rich medium supplemented or not with CFW at the indicated concentrations.
(TIF)

**S4 Fig. eIF2A genetically interacts with Xrn1.** Wild-type, *xrn1*-deg, *eif2aΔ* mutant and *xrn1-deg eif2aΔ* double mutant were cultivated in YPGlu medium until exponential-growth phase. IAA was added at a final concentration of 100 μM and CFW was added or not at a final concentration of 500 μg/ml. $OD_{600nm}$ was taken at indicated times. Error bars indicate the standard deviations of averages for at least three independent experiments.
(TIF)

**S1 Table. List of the *S. cerevisiae* strains used in this study.**
(DOCX)

**S2 Table. List of the plasmids used in this study.**
(DOCX)

**S3 Table. List of the oligonucleotides used in this study.**
(DOCX)

**S4 Table. List of the antibodies used in this study.**
(DOCX)

**S5 Table. List of the eIF2A mRNA targets.** The functions are provided by SGD https://www.yeastgenome.org/.
(XLSX)

**S1 Dataset. RIP-seq analysis.**
(XLSX)

**S2 Dataset. Transcriptome results.**
(XLSX)

**S3 Dataset. Mass spectrometry analysis.**
(XLSX)

**S1 File. S tables and references for strains used in this study.**
(DOCX)

**S1 Raw images.**
(PDF)

## Acknowledgments

We are grateful to the proteomics platform of the Pasteur Institute for the availability of the Orbitrap Q Exactive Plus mass spectrometer and the transcriptomic platform for the sequencing experiments. We thank Elodie Zhang and Frank Feuerbach for providing the Xrn1 mutant strains. We thank Cosmin Saveanu for discussions and criticisms on the manuscript and Madison Lenormand for polishing our english. We are grateful to Lucia Oreus for providing media and buffers used in this study and Christelle Lenormand for her administrative assistance.

## Author Contributions

**Conceptualization:** Laura Meyer, Baptiste Courtin, Abdelkader Namane, Alain Jacquier, Micheline Fromont-Racine.

**Data curation:** Laura Meyer, Baptiste Courtin, Maïté Gomard, Abdelkader Namane, Emmanuelle Permal, Alain Jacquier, Micheline Fromont-Racine.

**Formal analysis:** Laura Meyer, Baptiste Courtin, Maïté Gomard, Abdelkader Namane, Emmanuelle Permal, Gwenael Badis, Alain Jacquier, Micheline Fromont-Racine.

**Funding acquisition:** Alain Jacquier, Micheline Fromont-Racine.

**Investigation:** Abdelkader Namane, Alain Jacquier, Micheline Fromont-Racine.

**Methodology:** Laura Meyer, Baptiste Courtin, Maïté Gomard, Abdelkader Namane, Alain Jacquier, Micheline Fromont-Racine.

**Project administration:** Alain Jacquier, Micheline Fromont-Racine.

**Resources:** Alain Jacquier, Micheline Fromont-Racine.

**Software:** Emmanuelle Permal.

**Supervision:** Alain Jacquier, Micheline Fromont-Racine.

**Validation:** Laura Meyer, Baptiste Courtin, Maïté Gomard, Abdelkader Namane, Emmanuelle Permal, Alain Jacquier, Micheline Fromont-Racine.

**Visualization:** Laura Meyer, Baptiste Courtin, Maïté Gomard, Abdelkader Namane, Emmanuelle Permal, Alain Jacquier, Micheline Fromont-Racine.

**Writing – original draft:** Laura Meyer, Abdelkader Namane, Micheline Fromont-Racine.

**Writing – review & editing:** Laura Meyer, Alain Jacquier, Micheline Fromont-Racine.

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
