## [Decision Letter · Decision Letter 0]

2 Aug 2023

PONE-D-23-20160eIF2A represses cell wall biogenesis gene expression in Saccharomyces cerevisiaePLOS ONE

Dear Dr. Fromont-Racine,

Thank you for submitting your manuscript to PLOS ONE. After careful consideration, we feel that it has merit but does not fully meet PLOS ONE’s publication criteria as it currently stands. Therefore, we invite you to submit a revised version of the manuscript that addresses the points raised during the review process.

As you can see in the attached comments, reviewers raised several concerns that would need to be addressed before we could consider the study for publications. Especially, concerns about data availability, quality  and analysis should be addressed throughly. I also agree with the reviewer that the data need to support the conclusions. Thus, additional experiments will be needed in order to addressed the role of the cell wall integrity pathway in the phenotypes observed.

We look forward to receiving your revised manuscript.

Kind regards,

Patrick Lajoie, PhD

Academic Editor

PLOS ONE

Journal Requirements:

2. Thank you for submitting the above manuscript to PLOS ONE. During our internal evaluation of the manuscript, we found significant text overlap between your submission and previous work in the [introduction, conclusion, etc.].

Please revise the manuscript to rephrase the duplicated text, cite your sources, and provide details as to how the current manuscript advances on previous work. Please note that further consideration is dependent on the submission of a manuscript that addresses these concerns about the overlap in text with published work.

[If the overlap is with the authors’ own works: Moreover, upon submission, authors must confirm that the manuscript, or any related manuscript, is not currently under consideration or accepted elsewhere. If related work has been submitted to PLOS ONE or elsewhere, authors must include a copy with the submitted article. Reviewers will be asked to comment on the overlap between related submissions (http://journals.plos.org/plosone/s/submission-guidelines#loc-related-manuscripts).]

We will carefully review your manuscript upon resubmission and further consideration of the manuscript is dependent on the text overlap being addressed in full. Please ensure that your revision is thorough as failure to address the concerns to our satisfaction may result in your submission not being considered further.

"-MFR, AN and AJ (ANR-17-CE11-0049-01, ANR-17-CE12-0024-02)

-AN and AJ (ANR-18-CE11-0003-04)

Grants from the Agence Nationale de la Recherche, Institut Pasteur and the Centre National de la Recherche Scientifique.

LM and BC were supported by ANR-17-CE11-0049-01, ANR-17-CE12-0024-02 and ANR-18-CE11-0003-04."

Reviewers' comments:

Reviewer's Responses to Questions

**Comments to the Author**

1. Is the manuscript technically sound, and do the data support the conclusions?

Reviewer #1: No

Reviewer #2: Partly

2. Has the statistical analysis been performed appropriately and rigorously? 

Reviewer #1: No

Reviewer #2: No

3. Have the authors made all data underlying the findings in their manuscript fully available?

Reviewer #1: No

Reviewer #2: No

4. Is the manuscript presented in an intelligible fashion and written in standard English?

Reviewer #1: Yes

Reviewer #2: Yes

5. Review Comments to the Author

Reviewer #1: In this study the authors set out to understand the function of yeast eIF2A. To achieve this goal, they perform RIP-Seq. They identified numerous mRNAs attached to eIF2A including several cell wall genes. They explore the effects of overexpressing and removing eIF2a on yeast and the levels of cell wall genes. The follow up with IP-MS for eIF2A identifying Ssd1 as an interactor.

Overall, while there are some intriguing data from the -omic experiments, there are major issues with the study as presented:

1) Fig 1: Where is the full data for this? Significant RNAs needed to be listed in table form in the supplemental data as well as loaded to a proper repository.

2) Fig 1: Why are none of the other hits discussed? What other RNAs were detected? It’s hard to assess the importance of the selected genes without seeing these.

3) Fig 2: The standard cell wall phenotypes are temperature-sensitivity that can be suppressed by 1M sorbitol, caffeine and SDS sensitivity. The authors need to repeat their serial dilutions with these conditions, especially as their current phenotype is mild.

4) Fig3: The western blot quality is poor here especially Ccw14, so quantitation cannot be performed. Repeats need to be performed to give error bars and significance values for the graph

5) Fig 5: Same as Fig 1. Raw data needs to be loaded to PRIDE database with an accession number. The authors need to discuss all the other hits of proteins they found associated with the ribosome. Again, without this info it is very hard to assess if Ssd1 is of any major importance.

6) Fig 7: the quality of the blots are very poor here

7) Extensive studies by Levin and Molina groups have demonstrated that the majority of yeast cell wall genes are mediated by the cell integrity (PKC) pathway. The authors need to assess the phosphorylation status of Slt2 and induction of Rlm1 and Fks2 upon overexpression/silencing of eiF2 function.

8) Overall, biologically, it doesn’t make a lot of sense that deletion of eiF2A has such a minimal phenotype compared to overexpression.

9) The authors end the manuscript stating that this data may enable design of novel antifungal drugs. This seems to be reaching for a translational angle, when in reality there are already effective drugs that target the yeast cell wall and creating a drug to overexpress eif2a doesn’t seem very likely.

Reviewer #2: In this manuscript, Meyer et al. uncover a role for eIF2A in controlling expression of cell wall biogenesis genes at the translational level. RIP-seq experiments identify interactions between eIF2A and transcripts of cell wall biogenesis genes and the authors find deletion of the eIF2A gene leads to calcofluor white sensitivity. They show overexpression of eIF2A leads to decreased levels of cell wall proteins. Using AP-MS they find eIF2A associates with the translational repressor Ssd1 and discover that in ssd1∆ cells, eIF2A no longer interacts with some cell wall transcripts. They propose that eIF2A is a negative regulator of cell wall biogenesis genes in yeast. While it would have been nice to see additional experiments with the eif2a∆ strain to support their hypotheses, the authors present diverse experiments mainly using eIF2A overexpression that support their conclusions. However, there are some inconsistencies and key controls/data missing that must be addressed.

Major Points

1. The methods describing the mass spectrometry experiments are missing from the main text. They are listed in the supplemental, but this is a key experiment in the paper and the methods should be included in the main text. Additionally, it is not clear what label free quantification method was applied. The supplemental methods seem to imply it was an intensity based metric (supplemental line 64-65; “Protein group LFQ intensities were log2 transformed.”) but the legend of Fig 5B says “The x-axis represents the log2 fold change of each proteins between the average number of digested peptides…” suggesting a spectral counting based method. This must be clarified, ideally in the main text methods section.

2. Raw mass spectrometry data has not been deposited in an accessible repository like PRIDE or MassIVE where it can be analyzed by the public (and the reviewers of this work).

3. On line 407, the authors state that enriched proteins from the AP-MS experiment were determined based on the number of digested peptides identified in the eIF2A-TAP pull down relative to peptides identified in the whole cell lysate. However, background contaminants can be enriched independent of the tagged protein and therefore to determine enrichment, eIF2A-TAP purified samples must be compared to the proteins identified from a purification from a strain lacking eIF2A-TAP. From the supplemental MS methods, it looks like there was a set of negative control samples. This comparison (enrichment compared to control) should be the data presented in Figure 5B.

4. Similarly, the enrichment from the RIP-seq experiment was determined based on log2 fold change between the reads in eIF2A associated fraction relative to total RNA fraction according to the legend of Figure 1A. To control for non-specific interactions during the pull down, enrichment must be determined relative to a control no bait sample.

5. In figure 2C and D, the authors show OE eIF2A induces a growth defect. However, in Figure S1B this phenotype is not present. The authors should verify their results or explain this inconsistency in the text.

6. Statistical tests are missing throughout the manuscript. For example, the conclusions drawn from Figure 3B and 7B would be more robust with multiple test corrected p-values added on the figure or in the text.

7. The plasmid system to overexpress eIF2A is not clearly described. The authors should state if this is a centromeric or multicopy plasmid. Additionally, it is not clear in many of the figure legends if doxycycline was added to induce expression (other than stating cells were grown in -URA medium and harvested after 5 hours of eIF2A overexpression or not – for example, see Figure 2B, 3, 4, and 7). This should be clarified in the methods and figure legends.

8. Throughout the manuscript, representative cropped western and northern blots are shown however the authors should include the full uncropped western/northern blots with molecular weight markers in the supplemental to support their conclusions that the bands they are indicating represent the proteins they are quantifying.

9. On line 429, the authors state that eIF2A-TAP pulls down Ssd1-HA, however the blot shown in supplemental Figure S2A is not very convincing. Could the authors show a longer exposure or a different blot that better supports this claim? Otherwise, they should comment on why this pull down does not agree with the reciprocal pull down shown in Figure 5C.

10. Throughout the figure legends, the authors state the data shown is often representative of three replicates (for example – spot plates in Fig 2, western blot in Figure 5C, northern blot in Figure 6). However, these additional data are not found in the supplemental. For data completeness, they should be included.

11. In the discussion on line 626, the authors propose the mRNAs targeted by eIF2A are more easily degraded by Xrn1 to tune cell wall gene expression. However, the transcriptomics did not identify any changes in transcript levels which you would expect if Xrn1 was degrading transcripts. The authors should clarify this hypothesis so it is consistent with the data presented.

Minor Points

1. On line 88, “translation-driven” does not need to be hyphenated.

2. Throughout the manuscript “wild-type” is capitalized and should not be.

3. In the methods section, there is no description of how proteins were transferred from the polyacrylamide gel to a membrane and no description of the western blotting protocol.

4. On line 286, the authors do GO term enrichment on enriched mRNAs however there is no description of how the GO enrichment was done in the methods. Additionally, adding associated p-values would be useful to support the conclusions.

5. On line 301 in Fig 1A legend, log10 should be -log10.

6. On line 348 and 337, it should be “western blot analysis” not analyzes.

7. In figure 3B and 7B, it would be helpful to show the points from each individual replicate on top of the bars to give the reader a sense of the variability in these assays and how many replicates were performed.

8. The MS data represented in Dataset S3 should be better organized. Specifically, understanding which data corresponds to each sample/replicate in the first page and cleaning up the “volcano” page to be more interpretable would be useful for other researchers and for reproducibility.

9. Figure 5C is missing a “-“ in the RNase column above the eIF2A-HA column.

10. In a few places throughout the manuscript, the authors refer to mRNAs as “messengers” (see for example line 476 and line 601). It would be more accurate to say mRNAs.

11. Sentence on line 486 is hard to understand “was no longer be observed”.

12. On line 615, the authors state the most enriched eIF2A target is Xrn1. I believe it should be “interactor” instead of “target” as the authors did not find that eIF2A targeted the Xrn1 transcript.

13. On line 626, “negatively translated by eIF2A” does not make sense. Perhaps the authors meant “negatively regulated”?

6. PLOS authors have the option to publish the peer review history of their article (what does this mean?). If published, this will include your full peer review and any attached files.

Reviewer #1: No

Reviewer #2: No

---

## [Author Response · Author response to Decision Letter 0]

22 Sep 2023

RESPONSE TO THE EDITOR

We are pleased to see that the referees and you think that our manuscript deserves a publication in PLOS ONE. Please find enclosed a new version, revised according to the questions raised by the referees. 

This includes a point-by-point discussion of the reviewers’ concerns. However, some requested experiments seem to be beyond the scope of this study, and we estimated that the amount of work required to answer certain points would go way beyond the allocated period for the revision of this manuscript.

We received a similarity report for our manuscript, and we were very surprised to be asked to rewrite certain parts, since as far as we understood the similarity report, it concerns mostly basic sentences, such as “affiliation” or “methods”…, which are inevitably redundant from a study to another. However, we rephrased the “duplicated text” but we don’t know how we can change the sentences from sources 43, 34, 45, 49, 11, 79, 68, 77, 22, 82, 83, 86, 64, 52, 11, 71, 85, 53, which are basic sentences.

As requested : 

• We updated the Data Availability section. We added the accession number for PRIDE.

• We stated the financial disclosure and the role of the funder in this study as you suggested in the Funding section.

• We modified 3 figures (Figure 3, 5 and 7) and we added related informations in the first page of the S3 Dataset. 

• We uploaded a pdf file containing uncropped gels for western and northern in “Supporting Information”.

With this revised version, we think that this manuscript is ready for a publication in PLOS ONE journal and would be of interest for the scientific community.

Sincerely yours

------------

RESPONSE TO THE REVIEWERS

Reviewer #1: 

1) Fig 1: Where is the full data for this? Significant RNAs needed to be listed in table form in the supplemental data as well as loaded to a proper repository.

The full analysed data related to Fig 1 were present in S1 Dataset, the 146 significantly enriched RNAs were listed in S5 Table and all the transcriptomic raw data were deposited on the ENA Dataset server with the accession number PRJEB63554, as indicated at the end of the manuscript.

2) Fig 1: Why are none of the other hits discussed? What other RNAs were detected? It’s hard to assess the importance of the selected genes without seeing these.

The reason other hits were not discussed is mostly that there was no common function linking the corresponding genes. In addition to the transcripts related to the cell wall, we discussed the mRNAs in the most represented functional categories, such as the endoplasmic reticulocyte, membrane, mitochondria, and cell cycle. Altogether, with the cell wall mRNAs, these mRNA represent more than 50% of the enriched mRNAs. The other 64 mRNAs are listed in the S5 Table. A GO term finder search with these 64 candidates failed to group together some of them in a specific component or function or process. Therefore, they are not discussed but a description of their function was indicated in S5 Table.

3) Fig 2: The standard cell wall phenotypes are temperature-sensitivity that can be suppressed by 1M sorbitol, caffeine and SDS sensitivity. The authors need to repeat their serial dilutions with these conditions, especially as their current phenotype is mild.

The deletion mutation has no effect, however in the presence of calcofluor white, the growth phenotype was clear and reproducible (Figure 2B and Figure 8). We agree with the reviewer that the effect of the deletion could be also tested in the presence of sorbitol 1M, caffeine or SDS. However, all these reagents also induce cellular stress, and the observed phenotype could be indirect. Calcofluor binds to the polysaccharides of the chitin and is specific to the cell wall pathway. Therefore, we estimated that calcofluor white is the best reagent to use to specifically test phenotypes related with changes in the metabolism of the cell wall.

4) Fig3: The western blot quality is poor here especially Ccw14, so quantitation cannot be performed. Repeats need to be performed to give error bars and significance values for the graph.

We added corrected p-values, as also requested by the reviewer 2, to consolidate the data. We added in the legend of the Figure 3 the sentence: "Statistical analysis was performed by using a t-test, with the following obtained p-values: p=0.00182; p=0.01818; p=0.01077; p=0.02992 for Tos1, Ccw14, Sun4 and Cln1 respectively. Asterisks indicate statistical significances (*: p-value ≤0.05, **: p-value ≤ 0.01)."

5) Fig 5: Same as Fig 1. Raw data needs to be loaded to PRIDE database with an accession number. The authors need to discuss all the other hits of proteins they found associated with the ribosome. Again, without this info it is very hard to assess if Ssd1 is of any major importance.

As the answer to the point 1, the full analysed data related to Fig 5 were present in the S3 Dataset. The raw data are now accessible in PRIDE. We added: "Raw mass spectrometry data were posted in the ProteomeXchange Consortium via the PRIDE repository. The accession number is PXD043985." We apologize for the delay in depositing the results.

6) Fig 7: the quality of the blots are very poor here

We agree that the quality of the blots could be improved but this does not prevent nor change the interpretation of the results and the conclusions.

7) Extensive studies by Levin and Molina groups have demonstrated that the majority of yeast cell wall genes are mediated by the cell integrity (PKC) pathway. The authors need to assess the phosphorylation status of Slt2 and induction of Rlm1 and Fks2 upon overexpression/silencing of eiF2 function.

We looked at the transcriptomic results upon eIF2A overexpression or deletion conditions. The RML1 mRNA is not present but we observed that the FKS2 mRNAs was unchanged. We estimated that the amount of work required to investigate the phosphorylation status of Slt2 would go way beyond the allocated period for the revision of this manuscript and we estimated that this result is not essential for the message of the manuscript.

8) Overall, biologically, it doesn’t make a lot of sense that deletion of eiF2A has such a minimal phenotype compared to overexpression.

We agree with the reviewer 1 that eIF2A deletion has a minimal growth phenotype. However, this phenotype is clearly enhanced in presence of calcofluor white which, is fully consistent with a role of eIF2A in cell wall metabolism. 

9) The authors end the manuscript stating that this data may enable design of novel antifungal drugs. This seems to be reaching for a translational angle, when in reality there are already effective drugs that target the yeast cell wall and creating a drug to overexpress eif2a doesn’t seem very likely.

We eliminated the sentence. “but taken together, we can propose the possibility that mRNAs negatively translated by eIF2A are more easily targeted by Xrn1 to ensure fine-tuning of cell wall-related genes expression.”

Reviewer #2: 

Major Points

1. The methods describing the mass spectrometry experiments are missing from the main text. They are listed in the supplemental, but this is a key experiment in the paper and the methods should be included in the main text. Additionally, it is not clear what label free quantification method was applied. The supplemental methods seem to imply it was an intensity-based metric (supplemental line 64-65; “Protein group LFQ intensities were log2 transformed.”) but the legend of Fig 5B says “The x-axis represents the log2 fold change of each proteins between the average number of digested peptides…” suggesting a spectral counting based method. This must be clarified, ideally in the main text methods section.

The method describing the mass spectrometry experiments is now included in the main text. It is now indicated that the relative label-free quantification of proteins is based on intensities. We did not use spectral counting. 

The legend of the figure 5B was modified. We added the following sentence to clarify: "Volcano plot showed proteins enriched by eIF2A-TAP identified by mass spectrometry (LC-MS/MS). The x-axis represents the log2 fold change of each protein in the eIF2A-TAP enrichment compared with the total cell lysate." 

We moved the MS method from the supplemental informations to the main text.

2. Raw mass spectrometry data has not been deposited in an accessible repository like PRIDE or MassIVE where it can be analyzed by the public (and the reviewers of this work).

The raw data are now accessible in PRIDE. We added the sentence: "Raw mass spectrometry data were posted in the ProteomeXchange Consortium via the PRIDE repository. The accession number is PXD043985." We apologize for the delay of depositing the results in PRIDE. The full analysed data related to Fig 5 were however already present in the S3 Dataset.

3. On line 407, the authors state that enriched proteins from the AP-MS experiment were determined based on the number of digested peptides identified in the eIF2A-TAP pull down relative to peptides identified in the whole cell lysate. However, background contaminants can be enriched independent of the tagged protein and therefore to determine enrichment, eIF2A-TAP purified samples must be compared to the proteins identified from a purification from a strain lacking eIF2A-TAP. From the supplemental MS methods, it looks like there was a set of negative control samples. This comparison (enrichment compared to control) should be the data presented in Figure 5B.

We have modified the legend of the Figure 5B as indicated above and specified in the “Materials and Methods” section that we used a Relative label-free quantification of proteins based on LFQ intensities and not on the number of digested peptides.

Control purifications using cell cultures for a strain not expressing the purification tag led to a very small number of identified and quantified proteins, all of them known to be highly abundant in yeast cells. Comparing purifications with such a control is problematic because for most of the co-purified proteins there is no corresponding signal in the control. This problem can be removed if using total protein as the basis to calculate enrichment, as the number of proteins quantified in a total extract is orders of magnitude larger than in the control purification samples. We agree with the reviewer that, if the control purification would contain a large variety of proteins, they could serve to calculate enrichment as well.

4. Similarly, the enrichment from the RIP-seq experiment was determined based on log2 fold change between the reads in eIF2A associated fraction relative to total RNA fraction according to the legend of Figure 1A. To control for non-specific interactions during the pull down, enrichment must be determined relative to a control no bait sample.

We agree with the reviewer that there are two possible controls used in RIP-seq experiments, either a “no Tag” sample or the “input” sample. In the article Li Y et al NAR 2013 vol 41 n°8, they compared the two different controls and they concluded that both are “interchangeable”. One of the disadvantages of using the “no Tag” control is the high number of cycles that must be applied compared to the number of cycles required for the RIP sample. This has the consequence to over amplify the background and introduce a bias compared to the RIP sample. Another article mentioned how and why it is important to use the input as control to eliminate the background from the true targets (Wheeler EC Wiley Interdiscip Rev RNA. 2018 Jan;9(1):e1436.). Therefore, we preferred to use the input as a control to be able to apply the same treatment to the libraries (same number of PCR cycles). However, we performed a control similar with what the reviewer suggested (Fig S2) by coIP experiment using a strain with TAP-tag (strain Ssd1-HA) and no background noise could be detected.

5. In figure 2C and D, the authors show OE eIF2A induces a growth defect. However, in Figure S1B this phenotype is not present. The authors should verify their results or explain this inconsistency in the text.

This experiment has been conducted to do a test of complementation. The cells were plated on YPGlu instead of -URA to limit the eIF2A induction and to avoid the toxicity of the overexpression.

The sentence “Note that to limit the eIF2A induction and consequently, to avoid the toxicity, the cells were plated on YPGlu instead on -URA.” has been added in the text.

6. Statistical tests are missing throughout the manuscript. For example, the conclusions drawn from Figure 3B and 7B would be more robust with multiple test corrected p-values added on the figure or in the text.

We added corrected p-values, as also requested by the reviewer 1, to consolidate the data. We added in the legend of the Figure 3B the sentence: "Statistical analysis was performed by using a t-test, with the following obtained p-values: p=0.00182; p=0.01818; p=0.01077; p=0.02992 for Tos1, Ccw14, Sun4 and Cln1 respectively. Asterisks indicate statistical significances (*: p-value ≤0.05, **: p-value ≤ 0.01)." 

We added in the legend of the Figure 7B the sentence: "Statistical analysis was performed using a t-test, p=0.0054." 

7. The plasmid system to overexpress eIF2A is not clearly described. The authors should state if this is a centromeric or multicopy plasmid. Additionally, it is not clear in many of the figure legends if doxycycline was added to induce expression (other than stating cells were grown in -URA medium and harvested after 5 hours of eIF2A overexpression or not – for example, see Figure 2B, 3, 4, and 7). This should be clarified in the methods and figure legends.

The plasmid system to overexpress eIF2A is a multicopy plasmid derived from pCM189 containing the Ptet-off promoter, with the expression of eIF2A or SSD1 blocked in presence of doxycycline. Therefore, in all the experiments testing the effect of the eIF2A overexpression, the precultures were performed in the presence of doxycycline, then the cells were washed in medium without doxycycline and finally, the culture was done for 5 hours in the absence of doxycycline as indicated in the Material and Methods section, in the “eIF2A or SSD1 overexpression” paragraph. We have now specified in the legend of the Figure 2 (the first time we performed the eIF2A overexpression) that "the precultures were done in the presence of doxycycline (Dox) to prevent the expression of eIF2A which is under the control of the Ptetoff and plated on -URA medium without Dox."

8. Throughout the manuscript, representative cropped western and northern blots are shown however the authors should include the full uncropped western/northern blots with molecular weight markers in the supplemental to support their conclusions that the bands they are indicating represent the proteins they are quantifying.

This work has been done by several people. It is extremely difficult to find all the raw data in the computer of people who have left the lab: Baptiste Courtin (engineer) and Laura Meyer (post-doc) left the lab in February 2022 and June 2023 respectively. Abdelkader Namane and Alain Jacquier are now retired, Emmanuel Permal works now in another laboratory. The laboratory will be closed in July 2024 because I will also be retired. Therefore, I cannot benefit of the help of my collaborators to be able to recover all the primary results.

I try to do my best. As example, I join a picture of the full uncropped annotated western of the Figure 5C and Figure S2 which was present in the lab book of first author of the manuscript, Laura Meyer. 

I also send western with a merged MW for the western blots. The MW for the proteins are Tos1-TAP= 48 kDa+25 kDa (Tag); Sun4= 43+25; Cln1 62+25, Ccw14= 23+25; G6PDH=57, Ssd1=140. For, the northern blot, the only molecular weight that we use are the ribosomal rRNAs, the 18S and the 25S, and we verified the relative position on the membrane of the tested mRNAs in relation to each other.

9. On line 429, the authors state that eIF2A-TAP pulls down Ssd1-HA, however the blot shown in supplemental Figure S2A is not very convincing. Could the authors show a longer exposure or a different blot that better supports this claim? Otherwise, they should comment on why this pull down does not agree with the reciprocal pull down shown in Figure 5C.

We agree with the referee that the blot presented in Fig S2A is not very convincing. This is the reason it was only present in the supplemental data. We estimated that even if the quality of the blot is poor, the results were consistent with the MS data. This experiment is only a confirmation of the MS data which are largely more informative. The discrepancy between Fig 5C and Fig S2A can be explained by the fact that Ssd1 is an unstable protein. When we used Ssd1-TAP as bait we probably selected the full protein. We can remove Fig S2 if it is bothersome.

10. Throughout the figure legends, the authors state the data shown is often representative of three replicates (for example – spot plates in Fig 2, western blot in Figure 5C, northern blot in Figure 6). However, these additional data are not found in the supplemental. For data completeness, they should be included.

The answer is the same as for the point 8 above. I propose to eliminate the sentence “representative of three replicates”.

11. In the discussion on line 626, the authors propose the mRNAs targeted by eIF2A are more easily degraded by Xrn1 to tune cell wall gene expression. However, the transcriptomics did not identify any changes in transcript levels which you would expect if Xrn1 was degrading transcripts. The authors should clarify this hypothesis, so it is consistent with the data presented.

We are here in the section “Discussion”, the only place where we can speculate. Although this hypothesis did not seem outlandish, in view of the results, we chose to eliminate it. 

Minor Points

1. On line 88, “translation-driven” does not need to be hyphenated.

This modification has been done.

2. Throughout the manuscript “wild-type” is capitalized and should not be.

This modification has been done.

3. In the methods section, there is no description of how proteins were transferred from the polyacrylamide gel to a membrane and no description of the western blotting protocol.

This modification has been done.

We added the sentence: "into a gradient polyacrylamide gel (NuPAGE 4-12% Bis-Tris Gel from Invitrogen) and transferred onto Positively Charged Nylon Membrane (BrightStarTM.Plus Invitrogen) using Trans-Blot turbo transfert system (Bio-Rad), according to the manufacturer’s protocol. The membrane was incubated in blocking buffer (PBST with 5% milk) for 1 hour and then incubated with the appropriate antibody and dilution (S4 Table) for 1 hour at room temperature. Proteins were detected using the clarity ECL substrate (Bio-Rad) and Gel Doc XR system (Bio-Rad) according to the manufacturer’s protocol."

4. On line 286, the authors do GO term enrichment on enriched mRNAs however there is no description of how the GO enrichment was done in the methods. Additionally, adding associated p-values would be useful to support the conclusions.

This modification has been done.

We added the sentence: "Gene ontology (GO) term enrichment analysis was done on the Saccharomyces Genome Database (SGD) site according to [34]. It revealed that 24% of the selected mRNAs encode proteins required for cell wall biogenesis with a p-value of 6.56 e-22."

5. On line 301 in Fig 1A legend, log10 should be -log10.

This modification has been done.

6. On line 348 and 337, it should be “western blot analysis” not analyzes.

This modification has been done.

7. In figure 3B and 7B, it would be helpful to show the points from each individual replicate on top of the bars to give the reader a sense of the variability in these assays and how many replicates were performed.

This modification has been done.

We added in the legend of the Figure 3B and 7B, the sentence: "The dots correspond to the value obtained for each individual replicate."

8. The MS data represented in Dataset S3 should be better organized. Specifically, understanding which data corresponds to each sample/replicate in the first page and cleaning up the “volcano” page to be more interpretable would be useful for other researchers and for reproducibility.

A and B correspond to two independent experiments. Each experiment contains enriched sample and control (input) sample. Each experiment has been done in triplicate named A_01, A_02 and A_03 and B_01, B_02 and B_03.

All experiments from A and from B were treated together and named AB.

We explained the significance of the samples in the first page of the excel file. 

9. Figure 5C is missing a “-“ in the RNase column above the eIF2A-HA column.

This modification has been done.

10. In a few places throughout the manuscript, the authors refer to mRNAs as “messengers” (see for example line 476 and line 601). It would be more accurate to say mRNAs.

This modification has been done.

We agree with the reviewer.

11. Sentence on line 486 is hard to understand “was no longer be observed”.

This modification has been done.

We replaced “was no longer be observed” by “abolished.”

12. On line 615, the authors state the most enriched eIF2A target is Xrn1. I believe it should be “interactor” instead of “target” as the authors did not find that eIF2A targeted the Xrn1 transcript.

This modification has been done. We agree with the reviewer.

13. On line 626, “negatively translated by eIF2A” does not make sense. Perhaps the authors meant “negatively regulated”?

We agree with the the reviewer. Anyway, this sentence was now removed.

---

## [Decision Letter · Decision Letter 1]

2 Oct 2023

PONE-D-23-20160R1eIF2A represses cell wall biogenesis gene expression in Saccharomyces cerevisiaePLOS ONE

Dear Dr. Fromont-Racine,

Thank you for submitting your manuscript to PLOS ONE. After careful consideration, we feel that it has merit but does not fully meet PLOS ONE’s publication criteria as it currently stands. Therefore, we invite you to submit a revised version of the manuscript that addresses the points raised during the review process.

As you can see the reviewer raised two minor points that should be easily addressed in the text. 

We look forward to receiving your revised manuscript.

Kind regards,

Patrick Lajoie, PhD

Academic Editor

PLOS ONE

Journal Requirements:

Reviewers' comments:

Reviewer's Responses to Questions

**Comments to the Author**

1. If the authors have adequately addressed your comments raised in a previous round of review and you feel that this manuscript is now acceptable for publication, you may indicate that here to bypass the “Comments to the Author” section, enter your conflict of interest statement in the “Confidential to Editor” section, and submit your "Accept" recommendation.

Reviewer #1: All comments have been addressed

Reviewer #2: (No Response)

2. Is the manuscript technically sound, and do the data support the conclusions?

Reviewer #1: (No Response)

Reviewer #2: Yes

3. Has the statistical analysis been performed appropriately and rigorously? 

Reviewer #1: (No Response)

Reviewer #2: No

4. Have the authors made all data underlying the findings in their manuscript fully available?

Reviewer #1: (No Response)

Reviewer #2: Yes

5. Is the manuscript presented in an intelligible fashion and written in standard English?

Reviewer #1: (No Response)

Reviewer #2: Yes

6. Review Comments to the Author

Reviewer #1: (No Response)

Reviewer #2: The authors have done a good job to address most of the reviewer comments and I believe the manuscript will be of interest to the field, especially now that the raw transcriptomic and proteomic data is available. However, there are still two concerns I have:

1. I appreciate that the authors have added statistical significance values to their data, however it is not clear which multiple hypothesis testing correction method was applied. This should be stated in the methods or figure legends.

2. For the mass spectrometry data, the authors say there is a “vast number of background binding proteins” in the methods on line 293 but then in the response letter they state “Control purifications using cell cultures for a strain not expressing the purification tag led to a very small number of identified and quantified proteins”. Relatedly, the methods on line 293 still say “replicates of affinity-enriched bait samples were compared to a set of negative control samples” which contradicts the response letter and figure which compare the pull down to input. To address these issues, I recommend the authors correct the methods to reflect the comparison to input whole proteome, state in the text how many proteins were identified in their control pull downs and for proteins of interest state the fold-enrichment in the control sample compared to input (or state if they were detected in the control pull down at all).

7. PLOS authors have the option to publish the peer review history of their article (what does this mean?). If published, this will include your full peer review and any attached files.

Reviewer #1: **Yes: **Andrew Truman

Reviewer #2: No

---

## [Author Response · Author response to Decision Letter 1]

5 Oct 2023

We are pleased to see that the referees and you think that our manuscript deserves a publication in PLOS ONE. Please find enclosed a second version, revised according to the questions raised by the reviewer 2. 

Point 1:

The statistical significance values calculated for Figures 3 and 7 do not need multiple hypothesis testing correction because they were applied to a small number of samples.

For the transcriptomic data, the classical multiple hypothesis testing correction method was done by DESeq2.

For the proteomic analysis, the multiple hypothesis testing correction method was done by the Perseus software. We added “implemented in Perseus” line 302 and together with the reference 36.

Point 2:

We agree with the Review 2 that there is a contradiction between the text and the rebuttal letter on this point.

We deleted this part of the sentence line 293 “in the presence of a vast number of background binding proteins”

As suggested by the reviewer 2, we added information about the number of proteins identified and quantified: line 433-435

A total number of 1509 proteins were identified in either input or purified samples but only 1037 were quantified in both. The enrichment level for each protein is indicated in the S3 Dataset.

Additional point:

We detected an error in the Material and methods related to the Western blot method. We replaced “Positively Charged Nylon Membrane (BrightStarTM.Plus Invitrogen)” by “Nitrocellulose Membranes 0.45µm (Bio-Rad)”

With this second revised version, we hope that this manuscript is now ready for a publication in PLOS ONE journal and would be of interest for the scientific community.

Sincerely yours

Micheline Fromont-Racine

---

## [Editor Report · Decision Letter 2]

10 Oct 2023

eIF2A represses cell wall biogenesis gene expression in Saccharomyces cerevisiae

PONE-D-23-20160R2

Dear Dr. Fromont-Racine,

We’re pleased to inform you that your manuscript has been judged scientifically suitable for publication and will be formally accepted for publication once it meets all outstanding technical requirements.

Kind regards,

Patrick Lajoie, PhD

Academic Editor

PLOS ONE
---

## [Editor Report · Acceptance letter]

13 Nov 2023

PONE-D-23-20160R2 

eIF2A represses cell wall biogenesis gene expression
in *Saccharomyces cerevisiae*

Dear Dr. Fromont-Racine:

I'm pleased to inform you that your manuscript has been deemed suitable for publication in PLOS ONE. Congratulations! Your manuscript is now with our production department. 

Kind regards, 

on behalf of

Dr. Patrick Lajoie 

Academic Editor

PLOS ONE